# Sample composition alters associations between age and brain structure

Kaja Z. LeWinn[1], Margaret A. Sheridan[2], Katherine M. Keyes[3], Ava Hamilton[3] & Katie A. McLaughlin [4]

Despite calls to incorporate population science into neuroimaging research, most studies recruit small, non-representative samples. Here, we examine whether sample composition influences age-related variation in global measurements of gray matter volume, thickness, and surface area. We apply sample weights to structural brain imaging data from a community-based sample of children aged 3–18 ($N = 1162$) to create a "weighted sample" that approximates the distribution of socioeconomic status, race/ethnicity, and sex in the U.S. Census. We compare associations between age and brain structure in this weighted sample to estimates from the original sample with no sample weights applied (i.e., unweighted). Compared to the unweighted sample, we observe earlier maturation of cortical and sub-cortical structures, and patterns of brain maturation that better reflect known developmental trajectories in the weighted sample. Our empirical demonstration of bias introduced by non-representative sampling in this neuroimaging cohort suggests that sample composition may influence understanding of fundamental neural processes.

[1] Department of Psychiatry, University of California, San Francisco, 401 Parnassus Ave., San Francisco 94143, USA. [2] Clinical Psychology Department, University of North Carolina at Chapel Hill, 235 E. Cameron Avenue, Chapel Hill, NC 27599, USA. [3] Department of Epidemiology, Columbia University, 722 West 168th Street #724, New York, NY 10032, USA. [4] Department of Psychology, University of Washington, Box 351525, Seattle, WA 98195, USA. Correspondence and requests for materials should be addressed to K.Z.L. (email: kaja.lewinn@ucsf.edu)

Most neuroimaging studies rely on relatively small samples that are not representative of a well-defined target population. This has resulted in multiple calls to incorporate population science approaches into neuroimaging research[1, 2]. To date, however, the impact of convenience sampling on neuroimaging findings has not been examined empirically. In the current study, we address this need by examining whether sample composition influences age-related variation in brain structure among children in the United States.

All participants in research studies are drawn from target populations, even if study investigators do not explicitly define or enumerate that population. Even when a target population is defined (e.g., adults between the ages of 25 and 40 in the United States), study participants are unlikely to represent that target population unless they are randomly selected. Decades of methodological work in epidemiology and population science has shed light on the conditions that limit generalizability of findings generated from such non-representative samples[3–5]. This work suggests that sample composition may influence a study's conclusions when the association between the independent and dependent variable (e.g., age and brain structure) differs between those selected into the study and those who are eligible from the target population but not included[6, 7]. Such a scenario is likely to occur when study participants do not represent the target population in characteristics known to influence neural structure or function, for example, socioeconomic status (SES)[8]. Participants recruited into neuroimaging studies are not typically selected to be representative of a known target population, under the assumption—implicit or explicit—that basic neural functions (e.g., visual processing) in healthy individuals are not influenced by sample characteristics. Study findings are often assumed to reflect universal aspects of brain structure and function regardless of the sampling strategy. However, this assumption is largely untested and likely false.

There are exceptional examples of neuroimaging studies that have attempted to select representative samples[9, 10]; however, logistical challenges and study design decisions reduce the generalizability of findings from these studies to the broader U.S. population. In the foundational NIH MRI Study of Normal Brain Development, investigators selected a sample representative of the population in the study areas[9]; however, this study included numerous exclusion criteria (e.g., the presence of clinically significant mental health symptoms) that reduced the true representativeness of the sample[2]. The more recent NKI Rockland study was also designed to minimize sampling bias and maximize generalizability and included a representative sample of children and adults from Rockland County, NY[10]. Although this study represents a considerable advance toward representative sampling in cognitive neuroscience, participants were from a single geographic location and had higher levels of SES than in the U.S. population overall, indicating that this sample does not fully represent the U.S. While sample composition has become a growing area of focus in neuroimaging research[1, 2], to date there are no neuroimaging studies based on a representative sample of the U.S. population.

Here, we test the hypothesis that the use of non-representative samples in neuroimaging studies may influence interpretation of the association between age and brain structure. Age-related variation in brain structure in childhood and adolescence has been examined frequently in cognitive neuroscience. Prior studies have demonstrated substantial heterogeneity in the pattern of developmental change across brain structures and in the age at which peak thickness and surface area are reached for different cortical regions[11–15]. In the current study, we use a large neuroimaging data set of typically developing children, the Pediatric Imaging, Neurocognition and Genetics (PING) study[16], to examine whether sample composition influences age–brain structure associations. We use 2010 U.S. Census data to estimate the national distributions of basic socio-demographic characteristics (i.e., race/ethnicity, age, sex, parental educational attainment, and income) for children in the age range of the study sample (3–18 years). We apply sample weights based on these distributions to the PING sample using a common epidemiological and survey method procedure called raking to create a weighted PING sample that approximates a representative sample

**Table 1 Demographic characteristics in the United States and in the PING sample**

| Socio-demographic variables | ACS (N = 58,806,391) | PING (N = 1162) | | | | |
|---|---|---|---|---|---|---|
| | | Unweighted | | | Weighted | |
| | % | % | % Difference (unweighted PING-ACS) | | % | % Difference (weighted PING-ACS) |
| *Race* | | | | | | |
| White | 69.9 | 42.3 | −27.6 | | 69.9 | 0.0 |
| Black | 13.6 | 10.4 | −3.2 | | 13.6 | 0.0 |
| Hispanic | 7.5 | 23.8 | 16.2 | | 7.5 | 0.0 |
| Other | 5.5 | 8.8 | 3.3 | | 5.5 | 0.0 |
| 2 + Races | 3.5 | 14.7 | 11.2 | | 3.5 | 0.0 |
| *Sex* | | | | | | |
| Male | 51.3 | 52.9 | 1.7 | | 51.3 | 0.0 |
| Female | 48.7 | 47.1 | −1.6 | | 48.7 | 0.0 |
| *Parental education* | | | | | | |
| HS or less | 37.2 | 13.8 | −23.5 | | 37.2 | 0.0 |
| Some college | 32.6 | 24.8 | −7.8 | | 32.6 | 0.0 |
| College degree | 18.9 | 26.7 | 7.8 | | 18.9 | 0.0 |
| More than college | 11.3 | 34.7 | 23.4 | | 11.3 | 0.0 |
| *Income* | | | | | | |
| <40 k | 33.5 | 24.8 | −8.7 | | 33.5 | 0.0 |
| 40–100 k | 40.5 | 37.3 | −3.3 | | 40.5 | 0.0 |
| ≥100 k | 26.0 | 37.9 | 12.0 | | 26.0 | 0.0 |

Note: This table includes the distributions of socio-demographic characteristics in (1) a representative sample of children in the United States aged 3–18 from the ACS 2009–2011, (2) the unweighted PING sample, (3) the weighted PING sample after the raking procedure was applied using the ACS distributions. We also show the differences in the distribution of these characteristics between the ACS and the unweighted and weighted PING samples

**Table 2 Best-fitting models for global measures of brain structure**

| | | | Unweighted PING data | | Weighted PING data | |
|---|---|---|---|---|---|---|
| | | | Beta (SE) | AIC | Beta (SE) | AIC |
| Total cortical volume | Linear | Age | −6051.98 (275.28) | 24177.19 | −6521.28 (284.88) | 25103.9 |
| | Quadratic | Age | −5832.79 (280.75) | **24167.07** | −6420.06 (284.38) | 25092.7 |
| | | Age$^2$ | −239.87 (68.71) | | −258.53 (70.85) | |
| | Cubic | Age | −6359.63 (622.73) | 24168.17 | −8017.43 (650.34) | **25087.2** |
| | | Age$^2$ | −253.34 (70.14) | | −275.03 (70.85) | |
| | | Age$^3$ | 16.07 (16.96) | | 48.11 (17.63) | |
| Total cortical thickness | Linear | Age | −0.029 (8.0E-4) | −1710.34 | −0.03 (8.0E-4) | −804.89 |
| | Quadratic | Age | −0.029 (8.0E-4) | −1715.10 | −0.03 (8.0E-4) | −803.05 |
| | | Age$^2$ | 0.001 (2.0E-4) | | 0 (2.0E-4) | |
| | Cubic | Age | −0.026 (1.8E-3) | **−1718.31** | −0.026 (1.9E-3) | **−806.79** |
| | | Age$^2$ | 0.001 (2.0E-4) | | 0 (2.0E-4) | |
| | | Age$^3$ | 0 (0.0E+0) | | 0 (1.0E-4) | |
| Total cortical surface area | Linear | Age | 600.1 (132.8) | 22706.62 | 375.23 (133.59) | 23568.14 |
| | Quadratic | Age | 806.14 (132.11) | **22653.52** | 443.56 (132.04) | 23537.98 |
| | | Age$^2$ | −242.64 (32.25) | | −187.26 (32.76) | |
| | Cubic | Age | 270.94 (294.85) | 22651.41 | −609 (301.55) | **23525.07** |
| | | Age$^2$ | −256.23 (32.87) | | −197.66 (32.63) | |
| | | Age$^3$ | 16.32 (8.04) | | 31.67 (8.17) | |
| Total sub-cortical volume | Linear | Age | 364.01 (26.76) | 19445.29 | 310.65 (26.61) | 20290.88 |
| | Quadratic | Age | 398.74 (27.02) | **19414.78** | 326.47 (26.24) | 20255.36 |
| | | Age$^2$ | −38 (6.61) | | −40.42 (6.54) | |
| | Cubic | Age | 387.39 (59.95) | 19416.74 | 85.78 (59.65) | **20237.46** |
| | | Age$^2$ | −38.29 (6.75) | | −42.91 (6.5) | |
| | | Age$^3$ | 0.35 (1.63) | | 7.25 (1.62) | |

Note: Beta estimates, standard errors (SE), and AIC fit statistics for linear, quadratic, and cubic models of age for total cortical volume, total mean cortical thickness, total cortical surface area, and total subcortical volume in both the unweighted and weighted PING data. The AIC of the best-fitting model for each outcome is in bold

of the U.S. To determine the impact of sample composition on age-related variation in brain structure, we compare associations of age with global and regional measures of gray matter structure in the original, unweighted PING sample (i.e., non-representative) to those from the weighted PING sample (i.e., more representative). We focus our analysis on global morphometric cortical gray matter measurements as well as measurements of each lobe of the brain. Specifically, we examine cortical volume, cortical surface area, and cortical thickness for the entire cortex and for the right and left hemispheres; we additionally examine cortical surface area and thickness of frontal, parietal, temporal, and occipital lobes. These are robust metrics of brain structure that are measured with high reliability relative to specific cortical regions[17]. We also examine the volume of three widely studied subcortical structures—amygdala, hippocampus, and basal ganglia—as well as total subcortical volume to determine whether sample composition has a greater influence on cortical vs. subcortical regions and on global vs. specific measures.

Our results suggest that sample composition alters the interpretation of how cortical and subcortical areas vary with age. In the weighted sample, we frequently observe cubic (S-shaped) developmental patterns for cortical surface area and subcortical volume and younger ages of peak surface area and volume compared to the unweighted sample. In contrast, we primarily observe quadratic (U-shaped) developmental trajectories and older ages at peak cortical surface area and subcortical volume in the unweighted sample. Our findings empirically demonstrate observable impacts of sample composition on cognitive neuroscience findings, even for questions about fundamental processes such as age-related change in neural structure.

## Results

**Demographic characteristics in PING and the U.S. population.** We estimated the distributions of categorical socio-demographic variables in the U.S. Census, the unweighted PING sample, and the weighted PING sample after applying a generalized, model-based raking procedure (Table 1). Raking is a sample weighting method that generates sample weights for each individual participant based on observed characteristics of the target population (in this case, children in the U.S.), so that the weighted sample better reflects the distributions of characteristics in the target population that are included in the weighting procedure[18, 19]. We used the distributions of race/ethnicity, sex, parental educational attainment, and income from the U.S. Census to weight the PING data so that our weighted sample better approximates a representative sample of children in the U.S. Compared to population totals derived from the U.S. Census for children aged 3–18 years, the unweighted PING sample had fewer participants of European Caucasian descent (42 vs. 70%) with a greater proportion of Hispanic (24 vs. 8%) and multiple race participants (15 vs. 4%; see Table 1). PING participants were also from higher SES families with a higher percentage of parents making $100,000/ year or more (38 vs. 26%) and with greater parental education (35 vs. 11% of parents with post college degrees). After weighting, there were no differences between the weighted PING sample and the U.S. population in the distributions of race/ethnicity, sex, parental education, or parental income. Because the same participants were included in both unweighted and weighted samples (i.e., the samples are not independent), distributions between these samples cannot be statistically compared[20]. However, the descriptive comparisons above demonstrate: (1) that there are socio-demographic differences between the unweighted PING sample and the U.S. population; and (2) that the weighted PING sample is similar to the U.S. population with regard to the socio-demographic characteristics included in the raking procedure.

**Estimating age-related variation in brain structure.** To examine the impact of sample composition on age-related change in

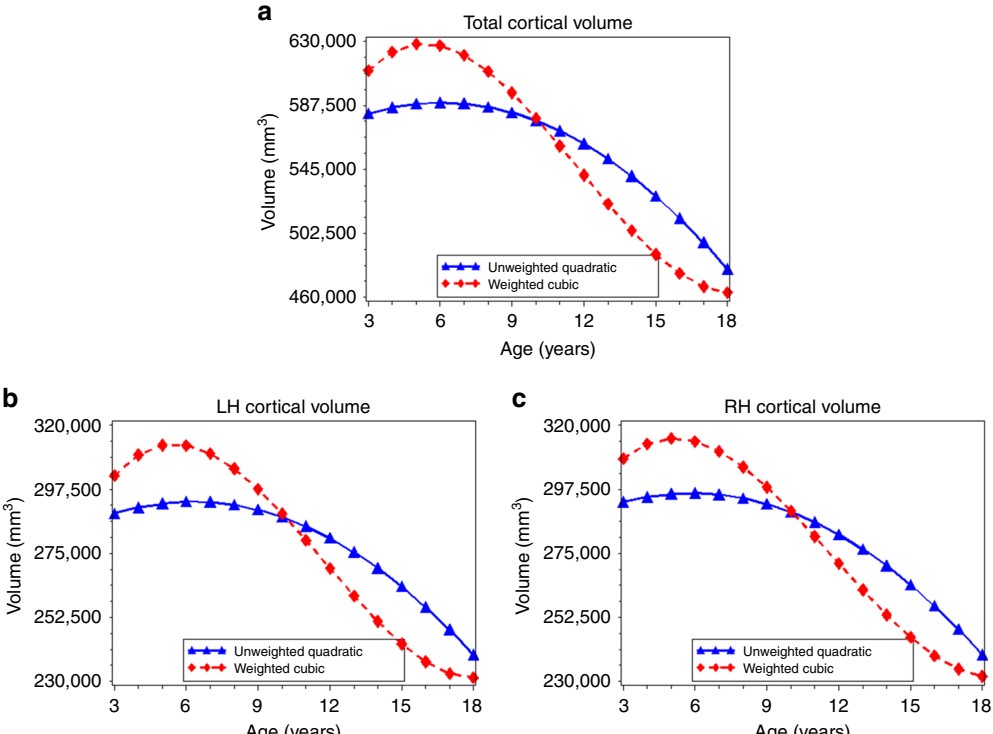

**Fig. 1** Age-related variation in cortical volume. Using the best-fitting regression models described in Table 2, we calculated the predicted values by age for total cortical volume (**a**), left hemispheric cortical volume (**b**), and right hemispheric cortical volume (**c**) in the unweighted (*blue lines*) and weighed (*red lines*) samples. LH left hemisphere, RH right hemisphere

neural structure, we used the Akaike information criteria (AIC) to identify the best-fitting model for the association between age and brain structure in both the unweighted and weighted PING sample. AIC provides a quantitative summary assessing how well a statistical model aligns with the underlying data compared with other models of the same data. It is commonly used for model selection (e.g., selecting covariates that provide the best fit to the data)[21], and is a standard method for determining model fit[22–24]. Consistent with prior investigations of age-related changes in brain structure[11–15], we compared different patterns of age-related variation by sequentially adding more complex polynomial terms for age into models of brain structure in the unweighted and weighted data. Including only age in the model assumes a linear relationship between age and brain structure; the addition of a quadratic term (i.e., age and age$^2$) generates a curved, U-shape relationship; and the addition of a cubic term (i.e., age, age$^2$, and age$^3$) generates the shape of an S-shaped sine wave. All three of these patterns have been observed in prior studies of age-related change in brain structure in children[11–15].

For each measurement of brain structure (total, right and left hemispheric cortical volume; total, right and left hemispheric cortical surface area; total, right and left mean hemispheric cortical thickness; frontal, occipital, temporal, and parietal lobe measures of cortical area and thickness; total subcortical volume, amygdala, hippocampal, and basal ganglia volume), we compared AIC between models that included age (linear); age and age$^2$ (quadratic); and age, age$^2$, and age$^3$ (cubic), and determined the lowest AIC across models in both the unweighted and weighted sample. If models that included more parameters than a linear parameter reduced AIC by at least 2.5 points, the more complex model was selected as a better fit to the data[25]. Throughout the results, the "best-fitting" model for the unweighted and weighted data refers to the model identified using this method. We report model fit statistics for all models examined (Table 2;

Supplementary Tables 2–5). To illustrate differences between the best-fitting unweighted and weighted models, we used parameter estimates from these models to generate predicted values for each brain structure metric by age and graphed these results (Figs. 1–5)[26]. Finally, we estimated the difference in age at peak area and volume between the unweighted and weighted models by calculating the first-order derivative of the fitted curves where appropriate (i.e., for quadratic and cubic models) (Table 3). These are common metrics derived from best-fitting regression models in developmental cognitive neuroscience[14, 27]. We describe differences in the best-fitting models, predicted values, and age at peak area and volume across the unweighted and weighted samples for each of these metrics (see "Methods" for full modeling approach).

**Age-related variation in cortical volume.** For total cortical volume (range: 382,286–753,561 mm$^3$), the best-fitting model included a quadratic age term in the unweighted data and a cubic age term in the weighted data (Table 2). For example, in the unweighted data a model that included both linear and quadratic age terms was a better fit for total cortical volume (AIC = 24167.07) compared to a model with only a linear age term (AIC = 24177.19); adding a cubic age term (AIC = 25087.24) did not improve model fit. In the weighted data, a model that included a linear, quadratic, and cubic age term (AIC = 25087.24) provided the best fit to the data when compared to a model including a linear term only (AIC = 25103.89) and a model with both a linear and quadratic term (AIC = 25092.66). For both right and left hemispheric cortical volume (left range: 189,847–375,672 mm$^3$; right range: 192,439–377,889 mm$^3$), the best-fitting models were also quadratic in the unweighted data and cubic in the weighted data (Supplementary Table 2).

Best-fitting regression models and predicted values suggested a higher peak cortical volume (unweighted: 589,414 mm$^3$,

**Table 3 Differences in age at peak volume and surface area**

|  | Age at peak (unweighted) | Age at peak (weighted) | Difference in age at peak (unweighted−weighted) |
|---|---|---|---|
| *Total cortical volume measures* |  |  |  |
| Total cortical volume | 6.1 | 5.3 | 0.8 |
| Total cortical volume (L) | 6.3 | 5.5 | 0.8 |
| Total cortical volume (R) | 5.9 | 5.1 | 0.8 |
| *Total and regional cortical surface area measures* |  |  |  |
| Total surface area | 12.1 | 9.7 | 2.4 |
| Total surface area (L) | 12.1 | 9.7 | 2.4 |
| Total surface area (R) | 12.1 | 9.7 | 2.4 |
| Frontal lobe (L) | 12.6 | 10.8 | 1.8 |
| Frontal lobe (R) | 12.5 | 11.0 | 1.5 |
| Occipital lobe (L) | 13.1 | 9.0 | 4.1 |
| Occipital lobe (R) | 12.3 | 8.8 | 3.5 |
| Temporal lobe (L) | 12.3 | 10.2 | 2.1 |
| Temporal lobe (R) | 12.7 | 10.7 | 2.0 |
| Parietal lobe (L) | 9.8 | 8.6 | 1.2 |
| Parietal lobe (R) | 9.7 | 8.6 | 1.1 |
| *Subcortical volume measures* |  |  |  |
| Total subcortical volume | 12.1 | 9.4 | 2.7 |
| Amygdala (bilateral) | 10.7 | 6.7 | 4.0 |
| Basal ganglia (bilateral) | 11.4 | 9.0 | 2.4 |
| Hippocampus (bilateral) | 10.0 | 8.3 | 1.7 |

Left hemisphere, Right hemisphere
Note: Thickness estimates not included because best-fitting models in unweighted and weighted models were effectively linear and declining over time

weighted: 628,575 mm$^3$), a more rapid decline in volume after age 6 (unweighted: $\beta_{age} = -5832.8$, $SE_1 = 281$, $\beta_{age^2} = -239.9$, $SE_2 = 69$; weighted: $\beta_{age} = -8017.4$, $SE_1 = 650$, $\beta_{age^2} = -275.0$, $SE_2 = 71$, $\beta_{age^3} = 48.1$, $SE_3 = 18$), and an age at peak cortical volume that was 0.8 years earlier in the weighted models as compared to the unweighted models (Table 3; Fig. 1a). For left and right hemispheric cortical volume, we observed a similar pattern and an age at peak volume that was also 0.8 years earlier in the weighted data compared to the unweighted data (Table 3; Fig. 1b, c).

**Age-related variation in cortical thickness**. The best-fitting models for total cortical thickness (range: 2.28–3.32 mm) included a cubic age term in both the unweighted and weighted data (unweighted: $\beta_{age} = -0.026$, $SE_1 = 0.002$, $\beta_{age^2} = 0.001$, $SE_2 = 0.0002$, $\beta_{age^3} = 0.0001$, $SE_3 = 0.00$; weighted: $\beta_{age} = -0.026$, $SE_1 = 0.002$, $\beta_{age^2} = 0.0001$, $SE_2 = 0.0002$ ($p = 0.55$), $\beta_{age^3} = -0.0001$, $SE_3 = 0.00$) (Table 2). Given the small parameter estimates associated with the quadratic and cubic age terms in the unweighted and weighted models, graphs of the predicted values indicate that these trajectories are effectively linear and decreasing with age (Fig. 2a). For example, a 1-year increase in age was associated with an approximately −0.029 change in total cortical thickness in both the unweighted and weighted data. Results for left (range: 2.30–3.32 mm) and right (range: 2.27–3.32 mm) hemispheric cortical thickness were similar to those for total thickness, with small parameter estimates for quadratic and cubic age terms, indicating an effectively linear decrease in thickness with increasing age in both the unweighted and weighted data (Fig. 2b, c; Supplementary Table 2).

Analyses of frontal, occipital, temporal, and parietal lobe thickness yielded several differences in the age terms included in the best-fitting models for the unweighted and weighted data (Supplementary Table 3). However, overall, patterns across lobes and between unweighted and weighted data were similar to what was observed with total cortical thickness, indicating primarily linear and decreasing associations between age and lobe-specific

measures of cortical thickness in both the unweighted and weighted data (Fig. 3a–h; Supplementary Table 3).

**Age-related variation in cortical surface area**. For total cortical surface area (range: 125,796–240,251 mm$^2$) the best-fitting model was quadratic in the unweighted data and cubic in the weighted data (unweighted: $\beta_{age} = 806.1$, $SE_1 = 132.1$, $\beta_{age^2} = -242.6$, $SE_2 = 32.2$; weighted: $\beta_{age} = -609.0$, $SE_1 = 301.6$, $\beta_{age^2} = -197.7$, $SE_2 = 32.6$, $\beta_{age^3} = 31.7$, $SE_3 = 8.2$) (Table 2). The graph of the predicted total surface by age (Fig. 2d) and age at peak surface area calculations (Table 3) demonstrate differences between the unweighted and weighted data, including an earlier age at peak total cortical surface area in the weighted model (9.7 years) compared to the unweighted model (12.1 years). For left and right hemispheric cortical surface area, best-fitting models were also quadratic in the unweighted data and cubic in the weighted data (Fig. 2e, f; Supplementary Table 2); we observed an age at peak volume that was also 2.4 years earlier in the weighted data compared to the unweighted data (Table 3).

For frontal, occipital, temporal, and parietal lobe surface area, we observed several differences in the best-fitting models across the unweighted and weighted data (Supplementary Table 4). For left and right frontal, occipital, and temporal lobes, the best-fitting model was quadratic in the unweighted data and cubic in the weighted data (Fig. 4a–h; Supplementary Table 4). First-order derivative calculations revealed that age at peak surface area for the occipital lobes occurred 4.1 (left hemisphere) and 3.5 (right hemisphere) years earlier in the weighted data than in the unweighted data (Table 3). The age a peak surface area was 2.0–2.1 years earlier for the right and left temporal lobes, and 1.5–1.8 years earlier for the right and left frontal lobes in the weighted compared to unweighted data (Table 3). For the left and right parietal lobe, the best-fitting model for both the unweighted and weighted data was cubic (Supplementary Table 4). However, both the predicted value graphs (Fig. 4e, f) and first-order derivative calculations (Table 3) indicated an earlier age at peak surface area (1.1–1.2 years) in the weighted data than in the unweighted data.

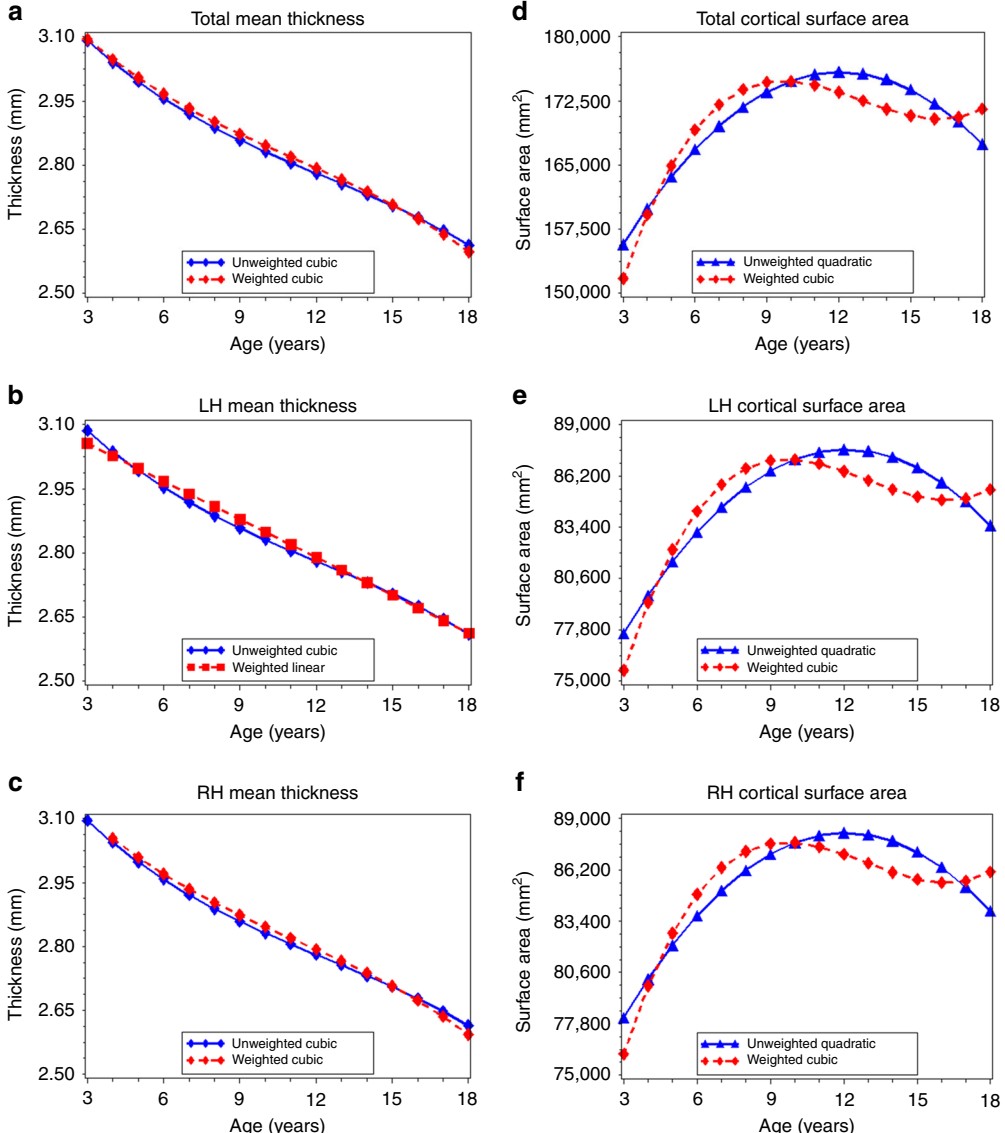

**Fig. 2** Age-related variation in cortical thickness and cortical surface area. Using the best-fitting regression models described in Table 2, we calculated the predicted values by age for total mean cortical thickness (**a**), left hemispheric cortical thickness (**b**), right hemispheric cortical thickness (**c**), total cortical surface area (**d**), left hemispheric cortical surface area (**e**), and right hemispheric cortical surface area (**f**) in the unweighted (*blue lines*) and weighed (*red lines*) samples. LH left hemisphere, RH right hemisphere

We also observed differences between the unweighted and weighted data in the relative age at peak surface area for each lobe of the brain. In the unweighted data, the frontal, occipital, and temporal lobes had similar ages at peak surface area (12.3–13.1 years), and only the parietal lobe reached peak area earlier in childhood (9.7–9.8 years). In contrast, the weighted data indicated an earlier age at peak area for the occipital and parietal lobes (between 8.6 and 9 years), followed by the temporal lobes (10.2–10.7 years), with frontal lobes maturing last (10.8–11.0 years).

**Age-related variation in subcortical volume**. Differences between unweighted and weighted data were also observed for total subcortical volume (range: 38,943–73,300 mm³). The best-fitting model was quadratic in unweighted data and cubic in weighted data (unweighted: $\beta_{age} = 398.7$, $SE_1 = 27.0$, $\beta_{age^2} = -38.0$, $SE_2 = 6.6$; weighted: $\beta_{age} = 85.8$, $SE_1 = 59.7$, $\beta_{age^2} = -42.9$, $SE_2 = 6.5$, $\beta_{age^3} = 7.3$, $SE_3 = 1.6$) (Table 2, Fig. 5a), with age at peak subcortical volume occurring 3 years earlier in

the weighted data (9.4 years vs. 12.1 years) (Table 3). For the amygdala and basal ganglia, differences between best-fitting models in unweighted and weighted data followed this same pattern (Fig. 5b, c; Supplementary Table 5). Age at peak volume was 4 years earlier for the amygdala and 2.4 years earlier for the basal ganglia in the weighted data compared to the unweighted data (Table 3). Though best described by a quadratic model in both the unweighted and weighted data, age at peak hippocampal volume followed a similar pattern as the other subcortical structures (Fig. 5d; Supplementary Table 5), with peak volume occurring 1.7–2.4 years earlier in the weighted compared to unweighted data (Table 3).

We note that in cubic models for both cortical surface area and subcortical volume, the tails of the distributions in the predicted models exhibit more variability than for linear and quadratic fits. For some outcomes, this resulted in a model that describes an increase in brain volume or surface area during late adolescence. This variability in the tails of the distribution is likely a function of sparse data and thus we caution inference from these models at

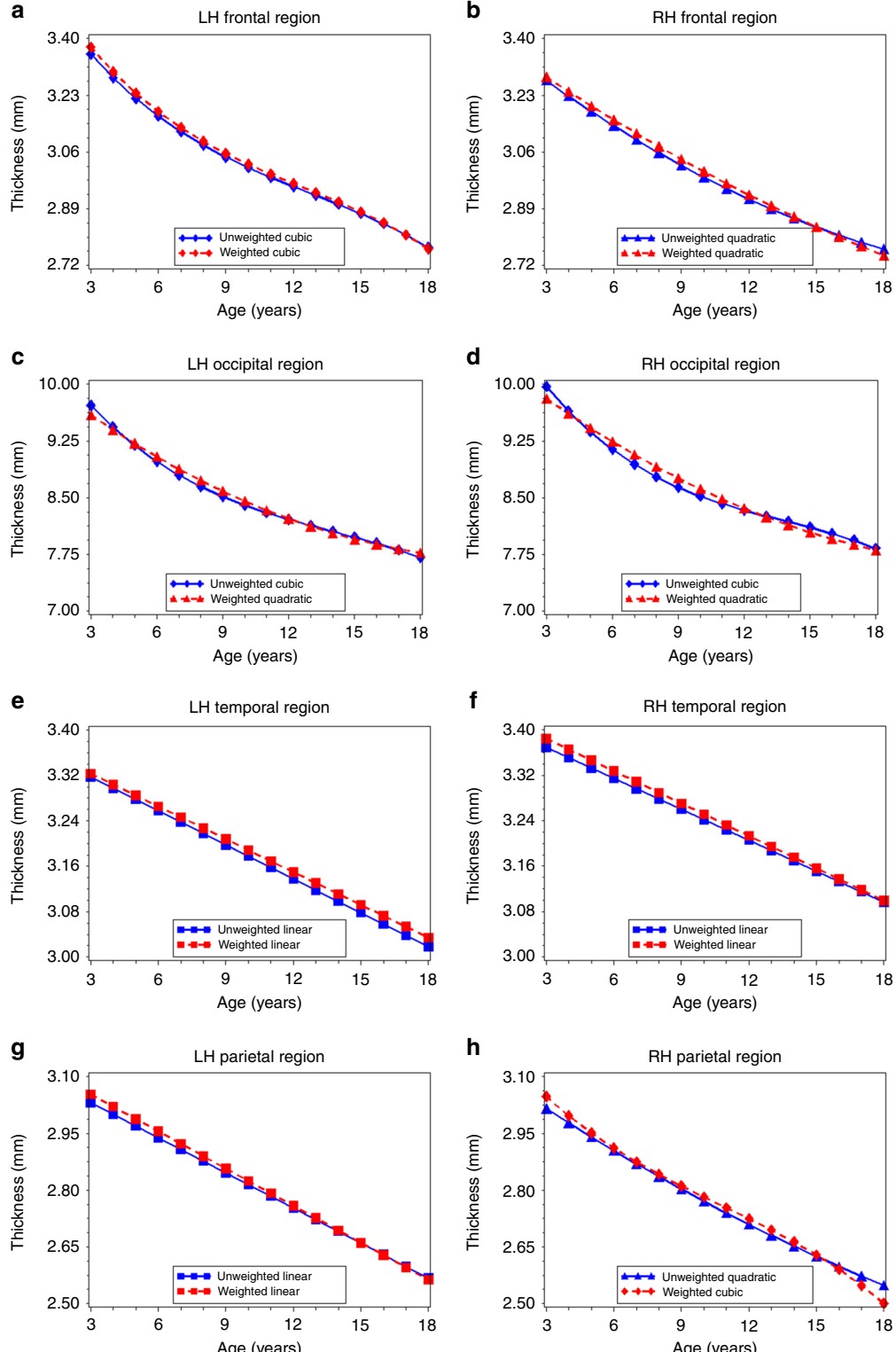

**Fig. 3** Age-related variation in cortical thickness for each lobe of the brain. Using the best-fitting regression models described in Table 2, we calculated the predicted values by age for: left (**a**) and right (**b**) frontal lobe thickness; left (**c**) and right (**d**) occipital lobe thickness; left (**e**) and right (**f**) temporal lobe thickness, and left (**g**) and right (**h**) temporal lobe thickness in the unweighted (*blue lines*) and weighted (*red lines*) samples. LH left hemisphere, RH right hemisphere

the tails (see Fig. 2d)[11]. In studies with wider age distributions, this upturn in late adolescence is not observed[11, 14]. Therefore, we did not interpret results from these highly variable tails of our age distribution.

## Discussion

Our findings suggest that current sampling practices in neuroimaging studies can produce systematic biases in our understanding of fundamental neural processes. We approximated a

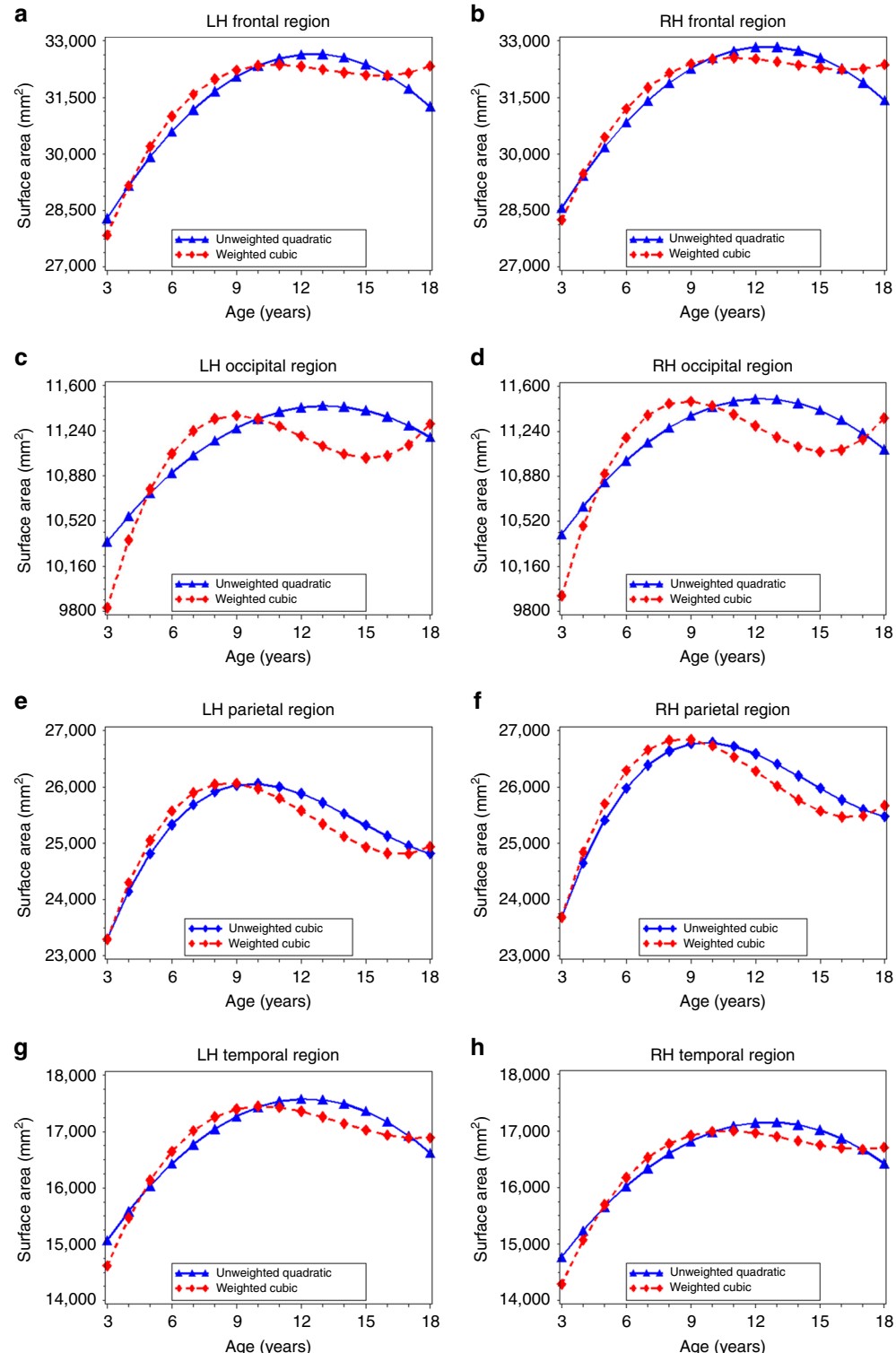

**Fig. 4** Age-related variation in cortical surface area for each lobe of the brain. Using the best-fitting regression models described in Table 2, we calculated the predicted values by age for: left (**a**) and right (**b**) frontal lobe surface area; left (**c**) and right (**d**) occipital lobe surface area; left (**e**) and right (**f**) temporal lobe surface area, and left (**g**) and right (**h**) temporal lobe surface area in the unweighted (*blue lines*) and weighted (*red lines*) samples. LH, left hemisphere, RH, right hemisphere

representative sample of U.S. children by applying a commonly used epidemiologic method of sample weighting to a large, community-based sample of typically developing children and estimated associations of age with global and regional measures of gray matter structure. We then compared these estimates to estimates of age-related variation derived from the original, unweighted study sample. These comparisons revealed differences between the unweighted and weighted sample on important measures of brain development such as age at peak area and volume, and the pattern of age-related change. Differences in age-related variation between the unweighted and weighted sample were observed across all measures of cortical surface area, cortical

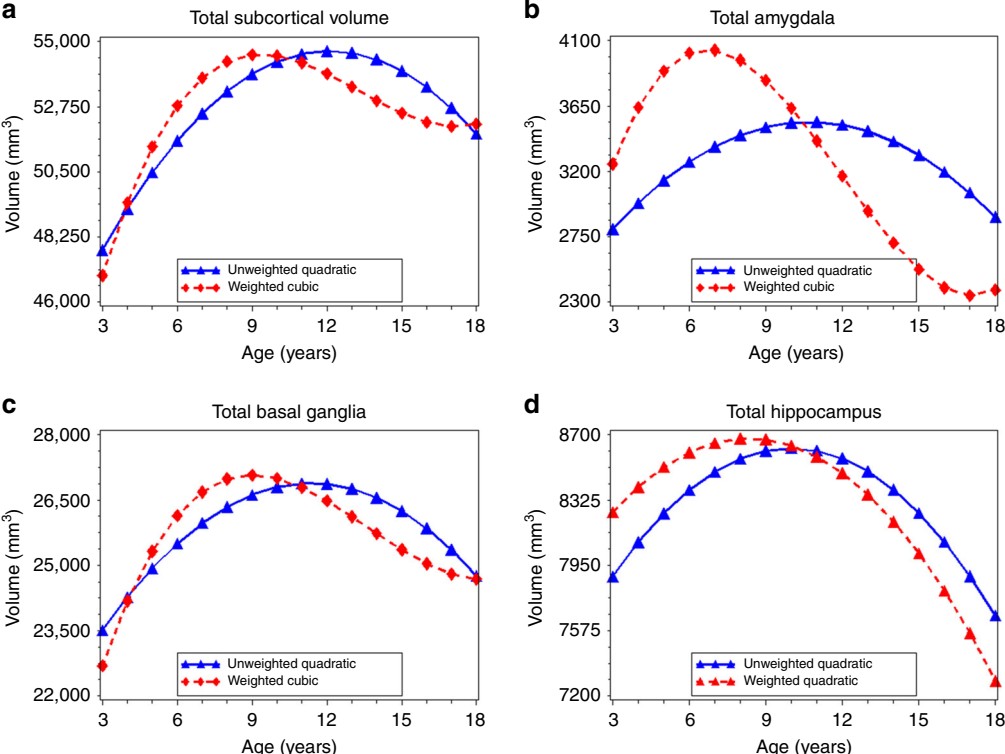

**Fig. 5** Age-related variation in subcortical volume. Using the best-fitting regression models described in Table 2, we calculated the predicted values by age for: total subcortical volume (**a**), total amygdala volume (**b**), total basal ganglia volume (**c**), and total hippocampal volume (**d**), in the unweighted (*blue lines*) and weighted (*red lines*) samples

volume, and subcortical volume but were more pronounced for lobe-specific measures of cortical surface area and for specific subcortical regions. Given that convenience sampling remains a common practice in neuroimaging research, our findings may have far-reaching implications for cognitive neuroscience studies.

Our descriptive results suggest that even large, community-based neuroimaging studies with multiple sites across the country will reflect the distributions of important socio-demographic characteristics of the U.S. population only if an explicit, representative sampling strategy is applied. This confirms what many neuroscience researchers suspect[1] but rarely examine: neuro-imaging studies implemented without an explicitly defined target population and appropriate sampling strategy rely on samples that differ in fundamental ways from the population of interest. In our study, PING participants had notably higher parental income and education and substantially different racial/ethnic composition than children of the same age in the U.S. population. Therefore, results from our unweighted models only generalize (or are transportable)[5] to the implicit target population of the PING study, which is undefined and has substantially different characteristics than the U.S. population as a whole. Although several innovative studies have attempted to generate more representative samples of smaller geographic regions[9, 10], this lack of generalizability is a fundamental characteristic of many cognitive neuroscience studies, which primarily rely on convenience samples[11–15].

Across measures of cortical surface area, cortical and subcortical volume, we observed a relatively consistent pattern of differences between the unweighted and weighted samples in estimates of age-related variation. The unweighted models were more likely to be quadratic, reflecting a gradual increase and then decline in lobe-specific surface area across age, whereas the weighted models were more likely to be cubic, reflecting a more

rapid increase in early/middle childhood followed by a tapering off in adolescence. This cubic pattern mirrors developmental change in other arenas (e.g., cognition) where rapid change is followed by periods of relative quiescence[28]. We also observed differences between the unweighted and weighted samples in the relative timing of cortical surface area development. In the unweighted data, the parietal lobe reached peak area first, followed by the temporal, occipital, and frontal lobes, which reached peak area around the same age. The pattern observed in the weighted data—with the occipital and parietal lobes reaching peak area first, followed by the temporal and then frontal lobes—is more consistent with what would be expected given the relative development of cognitive functions subserved by these regions[28–31], as well as evidence of brain development progression from sensory cortex to higher-order association cortex[32]. Similarly, while the unweighted data suggested an age at peak subcortical volume that occurred later than lobe-specific measures of cortical surface area, the weighted data indicated the opposite pattern with subcortical volume maturing earlier. Subcortical structures are thought to develop both phylogenetically and ontogenetically earlier than cortical regions, and are generally understood to subserve basic emotion and learning functions that are intact early in development and are critical for survival, such as fear and reward learning as well as other forms of explicit and implicit learning[33–35].

To further contextualize the impact of sample composition on age-related variation in brain structure in our study, we compare the magnitude of differences we observed in age at peak surface area between unweighted and weighted models to other studies of brain structure development. In a recent study of the relative timing of age at peak surface area across 84 regions of interest among youth aged 7–23 years, the earliest maturing regions reached peak surface area at age 8 compared to the latest

maturing regions, which reached peak area at age 11 (i.e., a 3 years difference between the earliest and latest maturing regions across the entire brain)[27]. We observed differences in age at peak surface area of a similar magnitude for the same measure of brain structure (e.g., surface area in the right occipital lobe) simply by applying sample weights to better reflect the demographics of the U.S. population. This suggests that sample composition may have a meaningful influence on metrics that are commonly derived from best-fitting regression models in developmental cognitive neuroscience.

Because the weighted data yielded a different interpretation of associations between age and brain structure, we can infer that these relationships were not uniform across the levels of socio-demographic characteristics included in our sample weighting procedure (i.e., child sex and race/ethnicity, parental education, parental income). The distributions of parental education and parental income in the original PING sample were substantially different from the U.S. Census distributions, and there is growing evidence that childhood SES is associated with brain structure, including cortical surface area[8], cortical thickness[8], and hippocampal volume[8, 36]. Furthermore, while there is limited research examining the extent to which SES modifies associations between age and brain structure, there is some evidence suggesting that low SES is associated with earlier and more rapid declines in frontal and temporal gray matter volume[37]. Indeed, the pattern of findings in our weighted data is consistent with an interpretation of earlier or faster brain maturation among low-SES children who were under-represented in the PING data; this is consistent with accumulating evidence that being raised in a resource-deprived environment accelerates development[38, 39]. However, it is important to note that our goal in the present analysis was not to attribute differences in the age–brain structure relationship to particular socio-demographic characteristics or to determine which characteristics are most important to include in weighting algorithms; rather, our findings highlight how sample composition across a number of basic socio-demographic characteristics may influence the answers obtained when examining fundamental questions in cognitive neuroscience and to whom those findings generalize.

One limitation of this study was that the PING sample was not representative of children in the U.S.[40]. In a true representative sample of children, the distribution of unmeasured characteristics would, on average, reflect that of the U.S. Our sample weights account for distributions of the measured characteristics of sex, race/ethnicity, parental education, and income, but not for all unmeasured characteristics that would render a sample representative of a target population. Characteristics that may further influence the age–brain structure association include birth weight[41], exposure to prenatal toxins, and exposure to traumatic violence[42]; however, these characteristics were not used to construct sample weights in the present study. Less frequently studied characteristics may also influence age–brain structure relationships. For example, PING participants were recruited from 10 urban study sites in 8 cities, leading to greater representation of children from urban and suburban areas and under-representation of children from rural locations. Future studies unable to implement a random sampling procedure but interested in post-stratification weighting may choose to assess additional participant characteristics (e.g., urban, suburban, rural) and use weighting techniques that include more complex algorithms to account for sample characteristics beyond basic socio-demographics. However, there is no substitute for true, representative sampling as traits endogenous to the participants that are too numerous to measure comprehensively may influence the likelihood of participation in samples of convenience (e.g., some participants may be more likely to hear of and participate in the study by word of mouth than others as a result of their social networks). As a result, although the weighted PING sample more closely resembles the U.S. population than the unweighted sample, it is not a truly representative sample.

The goal of our study was to demonstrate the potential impact of sample composition on fundamental relationships in cognitive neuroscience, not to provide the definitive answer regarding the association between age and brain structure for U.S. children. The PING study was well suited for this primary purpose given its very large sample size, appropriate age range to capture dynamic age-related differences in brain structure, and substantial geographic variability. The PING study remains the largest and most diverse neuroimaging study of U.S. children to date.

An alternative explanation of differences between the weighted and unweighted models is that they reflect differential motion artifact among lower-SES children who were weighted more heavily in the weighted sample to address their under-representation in the unweighted sample. Although it is certainly possible that small motion artifacts contribute somewhat to the variability across the weighted and unweighted samples, motion is unlikely to entirely explain the observed differences for several reasons. First, the PING study used a real-time motion correction algorithm during data acquisition[43, 44] and rigorous quality control procedures for the T1 data that dropped subjects with significant head motion. Second, if differential motion were driving these effects, we would expect that motion would be higher in the weighted sample where low-SES children had greater representation. This would produce a pattern of reduced cortical volume and thinner cortex in the weighted sample, particularly in the youngest children, as greater motion is associated with reductions in cortical thickness and volume[45, 46]. However, we find the opposite pattern. In the weighted data, the youngest children have greater cortical volume than the youngest children in the unweighted data, indicating that motion is an unlikely explanation for these findings.

We applied a parametric modeling strategy that allowed us to choose the best-fitting models for the unweighted and weighted sample by using a quantitative test of model fit (i.e., AIC). This approach was important for our demonstration because (1) it overlaps with standard modeling approaches used in developmental cognitive neuroscience[11, 13–15], and (2) it allowed us to quantitatively compare the best-fitting polynomial terms (e.g., linear, quadratic, cubic) across the unweighted and weighted data. There are many methods that are well suited for developmental cognitive neuroscience questions and could be employed to model age-related variation in brain structure (e.g., semi-parametric general additive models, local regression). We chose a commonly applied method of iterative polynomial term fitting where model fit estimates could be directly compared across weighted an unweighted data; future work should examine whether the application of sample weights to other methods of estimating age-related variation in brain structure produces similar differences in unweighted and weighted data.

Our results suggest that sample composition is likely to have a meaningful impact on cognitive neuroscience findings for many estimates and associations of interest; however, representative sampling is not feasible for all cognitive neuroscience studies. We make several suggestions that draw on population science principles to improve the reliability and generalizability of cognitive neuroscience findings. For small studies, we suggest that investigators follow the suggestions outlined by Falk et al.[1] to improve generalizability and comparisons across studies: (1) define the target population; (2) comprehensively report sampling and recruitment methodology; and (3) summarize the basic socio-demographic characteristics of the study sample (i.e., age, race and ethnicity, sex, and SES) at all stages of study implementation

(i.e., compare those recruited, those assessed, and those analyzed). In addition, a simple comparison of the distribution of socio-demographic characteristics in the analytic sample to that of the U.S. population or otherwise defined target population would also help clarify to whom findings generalize, and facilitate comparisons between studies using different samples.

For existing, large community-based studies that aim to address fundamental cognitive neuroscience questions, we suggest that, in addition to carefully describing the sampling strategy and socio-demographic characteristics of the sample, investigators consider applying a post-stratification weighting methodology to their final models as we did in the current study. Constructing sample weights is relatively straightforward and available in multiple statistical packages including SUDAAN, R, and SAS[47]. This would allow investigators to estimate the impact of sample composition on associations of interest. If differences between weighted and unweighted final models are small, one can have greater confidence in the generalizability of the results; if differences are large, associations could be reported based on a weighted sample that is more representative of the target population. It is important to note, however, that in addition to being large, community-based samples well suited for post-stratification weighting must include a diversity of participants that represent the broader characteristics of the target population, even if not in the same proportions as that target population. For example, it would not be possible to weight a sample including only non-Hispanic, white undergraduate students to represent all adults in the U.S. population. Further, if the sample has some diversity but there are few individuals with one of the characteristics included in the weighting algorithm, this may produce unreliable estimates of population-weighted totals. An example of a setting in which post-stratification methods could be applied successfully is the IMAGEN study[48], which was designed to examine genetic variation in brain and behavior, and, therefore, included participants that minimized ethnic diversity and maximized diversity in SES. However, no explicit representative sampling strategy was employed[48]. In studies such as this, applying a post-stratification weighting methodology to study questions that extend beyond the original intent and sampling strategy (e.g., questions unrelated to genetics)[49] will likely improve generalizability of those findings.

For future, large and costly neuroimaging studies aimed at understanding fundamental cognitive neuroscience questions, we do not recommend that weighting be used as a solution for non-representative sampling. Rather, we suggest the need for thoughtful sampling strategies that explicitly consider the target population of interest, and reflect an attempt to broadly recruit and represent such populations. An example of how this might be done in practice is observed in the National Institutes of Health recently funded Adolescent Brain and Cognitive Development (ABCD) study, which aims to assess brain development in over 10,000 adolescents from across the country. The ABCD study investigators intend to recruit a sample of boys and girls that represent the U.S. population of adolescents, and have developed a school-based recruitment strategy to accomplish this goal[50]. If conducted according to plan, this will be the first neuroimaging study that generates a true nationally representative sample, and will help ensure that the results of the study are generalizable to all adolescents in the United States. We suggest that this and future large-scale neuroscience initiatives include collaboration with population scientists (i.e., epidemiologists, demographers) to ensure that these sampling concerns are adequately addressed over the long term.

Finally, our study identifies a key issue that is important to consider regarding the replication of findings in human neuroscience research. Thus far, concerns about replication have primarily focused on the small sample sizes used in neuroimaging studies that are less likely to identify a true effect[51] or on the acquisition, processing, or analysis of structural or functional neuroimaging data, highlighting inconsistencies between approaches[52] and excessive false positive detection under some methods[53]. In this very large, well-powered study of typically developing children, we demonstrate that sample composition can also have a meaningful influence on the results one obtains, even when identical image preprocessing and model selection methods are applied. Because sample composition often varies widely between neuroimaging studies designed to answer similar questions (e.g., a sample of college undergraduates vs. a community sample of adults), sample composition is likely an additional contributor to the replication challenges facing cognitive neuroscience.

In summary, we find that the distribution of basic socio-demographic characteristics within a study sample, including race/ethnicity and SES, meaningfully influences the association between age and brain structure. Sample characteristics are likely to have even more relevance for studies aimed at understanding associations of the social and physical environment with brain structure or function, as many environmental characteristics vary significantly by race/ethnicity and/or SES (e.g., environmental toxin exposure, neighborhood quality, parental availability, exposure to violence)[54, 55]. Similarly, questions of interest to the social sciences regarding the neural processes underlying cognition, emotion, and behavior are also likely to be influenced by these fundamental social characteristics[56, 57]. Although there have been many calls to improve both the reliability and generalizability of neuroscience research[1, 2, 51], these calls have largely been ignored with regard to sampling strategies. Our findings demonstrate that the potential impact of sample composition on cognitive neuroscience research is not just theoretical, but in fact clearly observable in empirical work, even for questions about fundamental processes such as age-related change in neural structure.

## Methods

**Sample**. Data used in the preparation of this article were obtained from the PING study database (http://ping.chd.ucsd.edu/). The primary goal of PING has been to create a data resource of highly standardized and carefully curated magnetic resonance imaging (MRI) data, comprehensive genotyping data, and developmental and neuropsychological assessments for a large cohort of developing children aged 3–20 years. Participants were recruited through online and community-based advertising as well as word of mouth, in and around the ten PING data collection sites in eight cities (Los Angeles, Sacramento, New Haven, Boston, San Diego, Baltimore, New York, and Honolulu). Exclusion criteria included a history of neurological, psychiatric, medical, or developmental disorder. All participants gave informed consent for all study procedures; all parents provided consent and all child participants provided consent/assent as appropriate. Each data collection site's Institutional Review Board and Office of Human Subjects Research approved all procedures in this study.

A total of 1162 participants with data on neural structure who were age 3–18 years were included in our analytic sample. Parent participants reported on the level of educational attainment for themselves and their spouse. We used the highest reported education level in the home and created categories representing high school degree or less, some college, college degree, and more than college. Parents also reported on their total, yearly family income. This was divided into the following categories: <$40,000/year (less than 200% of the poverty line for a family of four in 2006), $40,000–$99,999/year (200–500% of the poverty line for a family of four), and $100,000/year or more (greater than 500% of the poverty line)[58]. These categories were chosen to represent meaningful levels of income[59] and also to allow us to adequately weight for the preponderance of high-income parents in the PING study. Parents also reported on the child's race, which was categorized as white, black, Hispanic, multiple race, or other.

**Image acquisition and processing**. The MRI protocol and standardized image processing techniques used in the PING study were designed to extract high-quality multimodal imaging data in a multisite study of children[40]. For each participant a single whole brain, T1 weighted structural magnetic resonance image was acquired in the sagittal plane using interleaved slice acquisition. All images

were acquired on a 3 T scanner at one of 10 different study sites using Siemens, GE, or Philips scanners. Acquisition parameters were standardized across sites and are detailed as follows: for Siemens: TE = 4.33 ms, TR = 2170 ms, flip angle = 7 degrees, 160 slices with $1 \times 1 \times 1.2$ mm voxels, FoV = 256; for Philips: TE = 3.1 ms, TR = 1665.9 ms, flip angle = 8 degrees, 170 slices with $1 \times 1 \times 1.2$ mm voxels, FoV = 256; GE: TE 1 = 4.0 ms, TR = 1500 ms, flip angle = 8 degrees, 170 slices with $1 \times 1 \times 1.2$ mm voxels, FoV = 256. To reduce motion, prospective motion correction (PROMO) was applied during acquisition[43]. Because different scanners are likely to have different field inhomogeneities resulting in differential sources of image distortion, a gradient field nonlinearity correction was applied prior to analysis[40].

Cortical thickness and surface area estimates were calculated with the FreeSurfer image analysis suite, which is documented and freely available for download online (http://surfer.nmr.mgh.harvard.edu). FreeSurfer morphometric procedures are well established[60–62], have demonstrated good test-retest reliability across scanner manufacturers and field strengths[63], have been validated against manual measurement[64, 65] and histological analysis[66], and have been successfully used in studies of children as young as age 4[67].

FreeSurfer methods applied to the processing of PING structural data included removal of non-brain tissue using a hybrid watershed/surface deformation procedure[68], automated Talairach transformation, previously validated in pediatric populations[69], and segmentation of the subcortical white matter and deep gray matter volumetric structures, separately validated for use with pediatric populations[67, 70]. FreeSurfer provided thickness and surface area estimates for 68 cortical regions (34 for each hemisphere), according to the Desikan-Killiany atlas[60, 71]. Labels for cortical gray matter were assigned using surface-based nonlinear registration to a gyral and sulcal-based atlas[62] and Bayesian classification rules[61, 71]. For subcortical structures, an automated, atlas-based, volumetric segmentation procedure was used to calculate volumes in mm³ for each structure, also executed in FreeSurfer[40].

Prior to inclusion in the final data set, neuroimaging data were required to pass rigorous quality-control procedures. All images were reviewed by trained technicians for significant motion artifacts, operator error and scanner dysfunction within 24 h of the scan to allow for the re-scanning of participants when possible[40]. T1-weighted images were examined slice by slice for any evidence of excessive motion and rated as either acceptable or for attempted rescan[40]. The subcortical segmentations, cortical parcellations, and white and pial surface reconstructions from the processed images were also reviewed by trained staff[40].

The publically available PING data set provides preprocessed, labeled, and quality controlled structural data for cortical surface area and thickness, and subcortical volumes based on the high-resolution T1-weighted scan. We chose to examine global and lobe-specific measures of cortical structure as they show high test-retest reliability and are more precisely estimated than smaller, individual structures[17]. Cortical gray matter measurements included total cortical volume, left/right hemispheric cortical volume, total subcortical volume, overall mean cortical thickness, left/right hemispheric mean cortical thickness, total cortical surface area, and left/right total cortical surface area. We also generated measurements for surface area and thickness for each lobe of the brain (frontal, occipital, temporal, and parietal) by combining regions identified in the Desikan-Killiany atlas (see Supplementary Table 1 for a complete list of regions)[60, 71]. We examined three subcortical structures—amygdala, hippocampus, and basal ganglia.

**Creating sample weights.** When a recruited sample does not adequately and proportionally cover segments of a target population, sample weights can be used so that the marginal totals of the adjusted weighted sample align with the target population on predefined characteristics (e.g., age, sex, race/ethnicity, SES, etc.). A classic way in which to create this alignment is through raking[18, 19]. In raking, the inverse of the marginal distribution of each variable to be included in the weight is iteratively multiplied across individuals in the sample. Each sample participant is then assigned a weight that is estimated as the difference between the unweighted value and the population distribution for the set of raked estimates. For illustrative purposes, consider two of the four variables we used for raking: sex and race. The raking procedure is essentially accomplished by first multiplying each individual by the inverse probability of being the sex that they are based on the overall population distribution of sex; the resulting estimates thus match the population distribution of sex, but not race. Then, each individual is multiplied by the inverse probability of being the race that they are given the overall population distribution of race. The resulting estimates thus match the population distribution of race, but now the sex estimates may not match population distributions. We then multiply again the individual by the inverse of the probability of their sex based on the population, and iteratively move through this sequence until there is convergence by which all of the weighted estimates match the population distributions within a caliper of error[18, 47]. The generalized raking procedure we followed was similar but with four variables: sex, race/ethnicity, parental education, and income, such that at the end of the procedure, the distributions of these demographic characteristics in the weighted sample were comparable to the population distribution of the U.S. Census in 2010. To improve the stability of estimates and ensure that results are not sensitive to a few individuals with extreme weights, it is traditional in raking procedures to "trim" the weights so that no extreme observation has undue influence[72]. We applied such trimming to our

sample, using an initial weight to estimate interquartile ranges (IQR) of the input sample and adjusted the weights so that no observation fell outside of 3 IQR of the initial weight.

We estimated population totals from the American Community Survey (ACS) Public Use Microdata Sample from 2009–2011. We then applied a raking procedure to the data using the "WTADJUST" procedure in SUDAAN, which employs a model-based approach and can be interpreted as a generalized raking procedure. The equations used to estimate the post-stratification weights are provided in the SUDAAN language manual[73], and we will summarize the main equation used for weight estimation here. Readers interested in full details of the generalized raking procedure are encouraged to refer to the manual for more details and full examples. We used the following equation for our post-stratification weight[73]:

$$\theta_k = \gamma_k \alpha_k = \gamma_k \left( \frac{l_k(u_k - c_k) + u_k(c_k - l_k)\exp\left(A_k x'_k \beta\right)}{(u_k - c_k) + (c_k - l_k)\exp\left(A_k x'_k \beta\right)} \right)$$

In this equation and as applied to our analysis, $k$ refers to each respondent in the PING data for which a final weight ($\theta_k$) was estimated. This final weight is a function of $\gamma_k$, the weight trimming factor used to stabilize the variance of the weighted estimates, and $\alpha_k$, the post-stratification adjustment. The post-stratification adjustment ($\alpha_k$) is described by a vector of the socio-demographic variables we included ($x'_k$, which in our model is sex, race/ethnicity, parental education, and income) and the model parameters ($\beta$) based on a logistic function. The remaining factors that determine the final weight are $A_k$, $l_k$, $u_k$, and $c_k$. These are all adjustments to improve weight stability, and include a lower bound ($l_k$), and upper bound ($u_k$), and a centering constant ($c_k$) for the weight of any individual in the data, which is required to be between the lower and upper board. $A_k$ is an additional constant that adjusts the final weight for stability. In summary, generalized raking procedures produce stable weight estimates based on a set of user-defined parameters that control the performance of the weight, as well as user-inputted variables that allow for the adjustment of each individual respondent so that the weighted sample as a whole is representative of the selected characteristics in the user-defined target population. We provide all of our statistical code as an online supplement that includes our user-defined parameters and assumptions that we made in the statistical model regarding weight trimming factors (see Supplementary Data 1).

**Regression models.** We next estimated separate models of the association of age with global and regional measures of gray matter structure to determine whether a linear, quadratic, or cubic term for age provided the best fit to the data. The best-fitting model for each measure was determined by comparing the AIC[21]. The more complex model (i.e., with quadratic or cubic terms) was selected when the AIC was at least 2.5 points lower than the AIC in for the less complex model[25]. AIC is commonly used for model selection (i.e., selecting covariates that provide the best fit to the data and selecting the best functional form of a model)[22–24]. Model fit statistics determine how well a particular model aligns with the underlying data, while taking into account the number of parameters in that model (rather than examining the statistical significance of each parameter individually). Model fit has long been accepted as the gold standard approach for model selection across a wide range of scientific disciplines, including the behavioral sciences and epidemiology[24, 25]; this approach is particularly well suited for deciding among models with polynomial terms[24].

All models included covariates for sex, race/ethnicity, parent educational attainment, family income, and scanner. Models for subcortical volume measurements also included intracranial volume (ICV). For both the unweighted and weighted samples, we used this same model building strategy to arrive at the best-fitting model to describe age-related variation in brain structure, so differences between the models can be attributed to the application of the sample weighting technique and underlying differences in the distribution of demographic characteristics in the unweighted and weighted samples.

To determine the extent to which differences in model parameterization led to meaningful differences in the interpretation of age-related variation between analytic approaches, we generated predicted values for each brain measure (area, thickness, and volume) at each age using the best-fitting unweighted and weighted data and graphed these results. We also calculated the difference in age at peak surface area and volume in both unweighted and weighted data where applicable (i.e., in quadratic and cubic models) by calculating the first-order derivative of the fitted curves. For quadratic models, we used the following formula to estimate peak age:

$$\text{MeanAge} + \frac{-a1\beta}{2 * a2\beta}$$

where MeanAge is the estimated sample mean, $a1\beta$ is the beta estimate for the linear age term in the regression model, and $a2\beta$ is the beta estimate for the age-squared term from the regression model. For cubic models, we used the following

formula to estimate peak age:

$$\text{MeanAge} + \frac{-(2*a2\beta) \pm \sqrt{(2*a2\beta)^2 - 4*(3*a3\beta)*a1\beta}}{2*(3*a3\beta)}$$

where MeanAge is the estimated sample mean, $a1\beta$ is the beta estimate for the linear age term in the regression model, $a2\beta$ is the beta estimate for the age-squared term from the regression model, and $a3\beta$ is the beta term for the age-cubed term from the regression model.

The predicted value graphs are intended to help readers visualize differences between the best-fitting unweighted and weighted data, as even models with quadratic or cubic terms can describe patterns of variation that are effectively linear. However, we are unable to compare aspects of these graphs (e.g., differences in slopes) with statistical tests because they are derived from different samples. For the same reason, calculations of age at peak surface area cannot be statistically compared between unweighted and weighted data and are included to provide a more tangible demonstration of how age-related trajectories in brain development may differ as a result of sample composition. For subcortical volume, final models also included ICV, and thus peak age was based on predicted values averaging the estimated volume within each 2-year age interval. To examine whether differences between unweighted and weighted models may be due to differences in head size, we examined subcortical ICV as an outcome. For subcortical ICV, the best-fitting models in the unweighted and weighted data were quadratic and indicated similar rates of change with age (see Supplementary Table 6 and Supplementary Fig. 1).

**Data availability**. The PING Data Resource includes neurodevelopmental histories, information about developing mental and emotional functions, multimodal brain imaging data, and genotypes for over 1000 children and adolescents. The data are available to members of the research community after submission of data use requests, agreement to the data use policies, and registration. More information about the PING Data Resource is available at http://pingstudy.ucsd.edu/ and http://ping.chd.ucsd.edu/. Our statistical code is available in Supplementary Data 1 and it is also available on GitHub at the following link: https://github.com/kajalewinn/PING.git.

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

## Acknowledgements

Funding for this project was provided by the National Institutes of Mental Health (K01MH097978 to K.Z.L.; R01-MH103291 and R01-106482 to K.A.M.), the National Institute on Drug Abuse (R03DA037405 to M.A.S.), the National Institute on Alcohol Abuse and Alcoholism (K01AA021511 to K.M.K.), an Early Career Research Fellowship from the Jacobs Foundation to K.A.M., and a Rising Star Research Award grant from AIM for Mental Health, a program of One Mind Institute (IMHRO) to K.A.M. Data collection and sharing for this project was funded by the Pediatric Imaging, Neuro-cognition and Genetics Study (PING) (National Institutes of Health Grant RC2DA029475). PING is funded by the National Institute on Drug Abuse and the Eunice Kennedy Shriver National Institute of Child Health & Human Development. PING data are disseminated by the PING Coordinating Center at the Center for Human Development, University of California, San Diego. We thank Dr. Randy Buckner for his helpful comments on an earlier version of this manuscript.

## Author contributions

K.Z.L. conceptualized the study. K.Z.L., K.A.M., and M.A.S. designed the study. K.M.K. developed the statistical methods and supervised A.H. who analyzed data and produced tables and figures. K.Z.L. and K.A.M. wrote and revised the manuscript; K.M.K. and M.A.S. wrote sections of the manuscript, contributed to interpretation of findings, and reviewed the manuscript.

## Additional information

**Competing interests:** The authors declare no competing financial interests.

