## [Peer Review File · Nature Communications]

Reviewers' comments:

Reviewer #1 (Remarks to the Author):

This groundbreaking paper has the potential to motivate change in the broad field of human neuroscience, particularly studies using neuroimaging. The authors have taken the large and important step of showing empirically that samples matter to conclusions in human neuroscience, even when answering the most basic questions in cognitive neuroscience. This work builds on studies in the social and behavioral sciences showing that sample composition (e.g., WEIRD samples, Henrich et al., 2010) affects even the most basic conclusions. This manuscript also directly answers the call from Falk et al., 2013, that neuroscience move to an approach more grounded in population science that considers the importance of sampling, samples, and generalizability. The innovation here is in using empirical data to concretely show how a lack of attention to sampling, even in the groundbreaking PING study, may lead to erroneous conclusions.

Major strengths of the study include a significant and important question, a novel approach, a relevant sample, thoughtful use of weighting procedures, and an examination of brain structure at multiple levels.

I have only a few minor points for the authors to consider as they polish this important study:

1. The authors took a sample (PING) that was already quite large with data from multiple sites. Thus, they took a relatively good sample to begin with and yet still show how a lack of sampling frame and attention to sample composition undermines basic conclusions. They then recommend a sampling weighting procedure as a potential remedy. Although this weighting approach may be appropriate for generating population estimates from the PING study, the authors may want to emphasize that few studies have this sort of data. That is, most neuroimaging studies are so far from a generalizable frame that weighting simply cannot lead to more accurate broader conclusions. For example, most cognitive neuroscience studies of undergraduate populations cannot be weighted to represent a population outside of the undergraduate institution.

I believe this point could be made clearer, as it has implications for our existing knowledge and future studies. I am a bit worried that some readers' take-away may be that weighting can solve these large issues in all studies, rather than that the most important take away is the need for more thoughtful sampling in study designs. The authors may want to offer the reader some suggestions of samples that could benefit from weighting versus those that simply cannot be generalized beyond a very narrow population. The authors may also want to comment on how sample size may influence the ability to use these weighting approaches.

2. Sampling and generalizability also have large implications for replication. Given the replication crises in neuroscience, psychology, and other related fields, it seems to be a missed opportunity not to point out that sampling may be one key issue in this crisis and

that better sampling may lead to more replicable conclusions.

3. Though the authors offer thoughtful limitations of their approach, there are many unknown attributes that may not be equated between the weighted PING sample and the US population. For example, I would guess that PING participants are more likely to be living in urban or suburban environments and were only sampled from specific portions of the country. Moreover, there are unmeasured personality variables influencing who may have been included in the sample given the recruitment methods (e.g., word of mouth). Thus, the authors may want to emphasize that without forethought in sampling, the use of weights cannot account for endogeneity. That is, if there wasn't a specific sampling frame in the first place, it is difficult to know what we don't know about how the current sample matches a very broad sample such as the US population.

4. Similarly, it would be helpful for the authors to explain to readers that if certain groups or attributes of individuals aren't represented (or minimally represented), then weights cannot lead to a full representation. For example, if PING does not contain groups such as Native American youth or youth who have been adjudicated to live in residential settings, then no amount of weighting can generate this information and the results cannot generalize to these individuals (and thus not the entire US population). Though this issue is common in any population study, it is important for authors to note what sampling frame is actually achieved (i.e., who is and is not likely to be represented). Additionally, a bit more information in the methods section is needed on how the US census data was used and combined with the ACS. For example, for variables such as SES, sex, and race, were figures used from the entire US population or only the population in the PING age range?

5. Finally, the point the authors make about the ABCD study sampling approach is relevant and timely. Of note, the current approach used by ABCD is focused on random sampling: <http://abcdstudy.org/school-selection.html>, though the study will need to stick to this approach to achieve the proposed sampling frame. The authors' point is important as this study is the perfect example of a study that will be much more impactful and valuable if the sampling frame is clear. That said, they may want to at least acknowledge that the current goals of ABCD are to use random sampling at the school level.

I applaud the authors for their efforts and hope this work can help push the field forward to a more generalizable and replicable understanding of human neuroscience. I appreciate the opportunity to contribute to reviewing this manuscript.

Signed: Luke W. Hyde, University of Michigan

Reviewer #2 (Remarks to the Author):

In "The representative developing brain: Does sampling strategy matter for neuroscience," LeWinn et al investigate the degree to which non-representative sampling may impact inference regarding normative patterns of brain development. This manuscript has a good

deal to recommend it, including an important topic, generally clear writing, and use of a large dataset. However, while I found this manuscript interesting, I felt that it left some aspects of this important problem under-explored. Suggestions for the authors to consider follow below:

1. Unpacking of results. The authors show that fitted models differ after weighted samples were constructed, but do not explain what drove those effects? Were some demographic effects more important to consider than others?

2. Data quality. Multiple studies (Reueter et al., Neuroimage 2015; Pardoe et al. Neuroimage 2016; Ducharme et al., Neuroimage 2016) have shown the impact of data quality on measures derived from structural images. I imagine that these concerns are as big if not bigger than those outlined in this paper—ignoring this factor in the weighting procedure seems like a potentially substantial flaw.

3. Datasets. There are increasing numbers of large-scale developmental datasets available. While PING is a good choice for this study, I was surprised that the authors did not use (or even reference) the NKI-Rockland sample as a counterpoint to PING, as it was explicitly designed with representative sampling in mind. Showing that after weighting the PING dataset became more like the rockland sample would increase confidence in the findings and provide a valuable replication.

4. Site effects. PING was a multi-site study, and site effects can be prominent in imaging data. Did the authors evaluate or control for these effects in their models?

5. Modeling. I was surprised the authors used polynomial models selected by AIC when the field has moved to data-driven semi-parametric models, such as general additive models; the PING consortium and others use primarily use this approach. I wonder if GAMs were to be used whether the same results would be found? Furthermore, rather than using low-dimensional summary measures like time at peak, I might prefer a quantitative model comparison over the entire age range—i.e., are the fitted values different only at peak, or at multiple age ranges?

Reviewer #3 (Remarks to the Author):

The paper examines the issue of non-representation in imaging studies in the neurocognitive literature. The authors find that there are some aspects of imaging that find little differences in a weighted, representative sample but in regional areas some quite important differences. The authors offer a novel use of weights to examine the potential bias that imaging studies may have due to lack of representing the population. This paper gives some evidence that the biases can be important and substantial.

The article is convincing. The only real weakness, and this is addressed by the authors, is that this study is not representative to begin with and so the estimates are of the potential bias. Also, the authors made reference to the ABCD 10,000 adolescent study and that study is using population sampling to ascertain the sample. That will be one of the first to really have the representation of both the brain and biology and will be available for researchers across the world to use.

The statistical analyses appears appropriate and relies on looking at differences in functional forms vs. mean differences. The reviewers have provided detail of their method and a link to their study and are providing the data. Thus, it should be straight forward to perform a replication of the data given the openness of the data and the procedures used in this study.

Overall, a good examination of the issue of lack of representation of neurocognitive studies in the field. It is also a call to action to consider the issue of universal processes when almost no studies have actually taken into account important differences in college samples and population samples.

Pamela Davis-Kean

Reviewer #4 (Remarks to the Author):

LeWinn and colleagues describe the impact of "representativeness" (or a lack off) on estimating trajectories of brain development in community-based samples.

This is a well-written manuscript with a very important message. I have only a few minor comments and suggestions.

In the second paragraph (around line 70 and line 83), you may wish to point out a census-based approach employed in the NIH MRI Study of Normal Brain Development. This is one of a few studies that attempted to do what the authors are calling (visi-a-vis demographic variables) for when recruiting children into an imaging study. The same study, however, introduced other biases – for example an extremely long list of exclusion criteria that likely resulted in a "supernormal" sample...

Around line 111, you may wish to draw attention to a recent paper by Ducharme (PMID:26463175) suggesting that non-linear relationships between cortical thickness and age may be "induced" by head motion.

On line 231, you may wish to add "created" after "children" to avoid a misunderstanding that this sample was representative to start with...

In the paragraph about the most pronounced differences between unweighted and weighted models in subcortical volumes (lines 288-300), you may wish to consider an alternative explanation: subcortical volumes are more difficult to estimate (than cortical volumes) and, possibly, more sensitive to imaging artifacts (such as head motion – see the Ducharme paper mentioned above). Could this be why the two models differ most?

When talking about calls "to incorporate population science approaches into neuroimaging research (line 48) and "into cognitive neuroscience" (line 228), you may wish to cite Paus 2010 (PMID:20496380) and 2013 (ISBN 978-3-642-36450-1).

RESPONSE TO REVIEWERS' COMMENTS

We thank all the reviewers for their thoughtful comments on our manuscript. We have carefully considered this feedback and have addressed each concern within the following response to reviewers (reviewers' comments are in italics, our responses are in normal font, and revised excerpts from our paper are in quotes) and in our revised manuscript. We believe this revised manuscript is much stronger as result of this review process. Thank you all for your time and consideration.

Reviewer #1 (Remarks to the Author):

This groundbreaking paper has the potential to motivate change in the broad field of human neuroscience, particularly studies using neuroimaging. The authors have taken the large and important step of showing empirically that samples matter to conclusions in human neuroscience, even when answering the most basic questions in cognitive neuroscience. This work builds on studies in the social and behavioral sciences showing that sample composition (e.g., WEIRD samples, Henrich et al., 2010) affects even the most basic conclusions. This manuscript also directly answers the call from Falk et al., 2013, that neuroscience move to an approach more grounded in population science that considers the importance of sampling, samples, and generalizability. The innovation here is in using empirical data to concretely show how a lack of attention to sampling, even in the groundbreaking PING study, may lead to erroneous conclusions.

Major strengths of the study include a significant and important question, a novel approach, a relevant sample, thoughtful use of weighting procedures, and an examination of brain structure at multiple levels.

I have only a few minor points for the authors to consider as they polish this important study:

1. The authors took a sample (PING) that was already quite large with data from multiple sites. Thus, they took a relatively good sample to begin with and yet still show how a lack of sampling frame and attention to sample composition undermines basic conclusions. They then recommend a sampling weighting procedure as a potential remedy. Although this weighting approach may be appropriate for generating population estimates from the PING study, the authors may want to emphasize that few studies have this sort of data. That is, most neuroimaging studies are so far from a generalizable frame that weighting simply cannot lead to more accurate broader conclusions. For example, most cognitive neuroscience studies of undergraduate populations cannot be weighted to represent a population outside of the undergraduate institution.

I believe this point could be made clearer, as it has implications for our existing knowledge and future studies. I am a bit worried that some readers' take-away may be that weighting can solve these large issues in all studies, rather than that the most important take away is the need for more thoughtful sampling in study designs. The authors may want to offer the reader some suggestions of samples that could benefit from weighting versus those that simply cannot be generalized beyond a very narrow population. The authors may also want to comment on how sample size may influence the ability to use these weighting approaches.

We thank the reviewer for this thoughtful comment, and agree that our central take-home message is not that weighting can solve the widespread issue of non-representative sampling in cognitive neuroscience. Rather, our goals were to demonstrate that sample composition could have meaningful influences on fundamental associations in cognitive neuroscience even in a very large community sample and to illustrate the potential challenges with inference in studies that do not explicitly outline a sampling strategy. We have made these points more explicitly in the discussion section of the revised paper.

Furthermore, although weighting may be a useful strategy to consider in samples that are large enough and have sufficient representation of key socio-demographic characteristics, we agree that weighting is not a solution for small samples or studies of limited representation (e.g., college student samples). Even neuroimaging studies with large and diverse samples may not be able to apply weighting strategies if data are not routinely collected on race/ethnicity, socio-economic status, and other socio-demographic characteristics, and smaller neuroimaging samples are likely to produce unreliable estimates of population weighted totals if there are few participants with a given characteristic. In summary, weighting is not a general solution to non-

representative sampling. We use it here to demonstrate that sample composition can have meaningful effects on inferences in cognitive neuroscience and highlight the need for more thoughtful sampling strategies that explicitly consider the target population of interest, and reflect an attempt to broadly recruit and represent such populations. We have included a more nuanced discussion of when it might be appropriate to consider post-stratification weighting throughout our revised discussion section.

Please see the first paragraph of our Limitations section (page 7-8) and second paragraph in the Implications and Conclusions section (page 9) for our expanded discussion of these issues.

2. Sampling and generalizability also have large implications for replication. Given the replication crises in neuroscience, psychology, and other related fields, it seems to be a missed opportunity not to point out that sampling may be one key issue in this crisis and that better sampling may lead to more replicable conclusions.

We agree with this point and thank the reviewer for drawing our attention to this additional implication of our study. We have added greater detail on this to the discussion section of the revised paper. Specifically, we note the following on page 10:

“Finally, our study adds to mounting concerns regarding the replication of findings in human neuroscience research. Thus far, these concerns have primarily focused on the small sample sizes used in neuroimaging studies that are less likely to identify a true effect⁴⁷ or on the acquisition, processing, or analysis of structural or functional neuroimaging data, highlighting inconsistencies between approaches⁴⁸ and excessive false positive detection under some methods⁴⁹. In this very large, well-powered study of typically developing children, we demonstrate that sample composition can have a meaningful influence on the results one obtains, even when identical image preprocessing and model selection methods are applied. Because sample composition often varies widely between neuroimaging studies designed to answer similar questions (e.g. a sample of college undergraduates versus a community sample of adults), sample composition is likely an additional contributor to the replication challenges facing cognitive neuroscience.”

3. Though the authors offer thoughtful limitations of their approach, there are many unknown attributes that may not be equated between the weighted PING sample and the US population. For example, I would guess that PING participants are more likely to be living in urban or suburban environments and were only sampled from specific portions of the country. Moreover, there are unmeasured personality variables influencing who may have been included in the sample given the recruitment methods (e.g., word of mouth). Thus, the authors may want to emphasize that without forethought in sampling, the use of weights cannot account for endogeneity. That is, if there wasn't a specific sampling frame in the first place, it is difficult to know what we don't know about how the current sample matches a very broad sample such as the US population.

We agree with the reviewer on this point, and have addressed this limitation in our discussion section on page 7:

“Less well-studied characteristics may also influence age-brain structure relationships. For example, PING participants were recruited from nine urban study sites, leading to greater representation of children from urban and suburban areas and under-representation of children from rural locations. Future studies unable to implement a random sampling procedure but interested in post-stratification weighting may choose to assess additional participant characteristics (e.g. urban, suburban, rural) and use weighting techniques that include more complex algorithms to account for sample characteristics beyond basic socio-demographics. However, there is no substitute for true, representative sampling as traits endogenous to the participants that are too numerous to measure comprehensively may influence the likelihood of participation in samples of convenience (e.g. some participants may be more likely to hear of and participate in the study by word of mouth than others as a result of their social networks). As a result, although the weighted PING sample more closely resembles the U.S. population than the unweighted sample, it is not a truly representative sample.”

4. Similarly, it would be helpful for the authors to explain to readers that if certain groups or attributes of individuals aren't represented (or minimally represented), then weights cannot lead to a full representation. For example, if PING does not contain groups such as Native American youth or youth who have been adjudicated

to live in residential settings, then no amount of weighting can generate this information and the results cannot generalize to these individuals (and thus not the entire US population). Though this issue is common in any population study, it is important for authors to note what sampling frame is actually achieved (i.e., who is and is not likely to be represented). Additionally, a bit more information in the methods section is needed on how the US census data was used and combined with the ACS. For example, for variables such as SES, sex, and race, were figures used from the entire US population or only the population in the PING age range?

We agree with the reviewer that no amount of weighting can account for a characteristic that is not present in the original data, and that weights are unlikely to be adequate if there are only a few participants with the selected characteristic. For example, it would not be possible to weight a sample of entirely non-Hispanic White children to be population representative of race/ethnicity. We have now more explicitly made this point in the discussion section (page 9):

“It is important to note, however, that in addition to being large, community-based samples well suited for post-stratification weighting must include a diversity of participants that represent the broader characteristics of the target population, even if not in the same proportions as that target population. For example, it would not be possible to weight a sample including only non-Hispanic, white undergraduate students to represent all adults in the U.S. population. Further, if the sample has some diversity but there are few individuals with one of the characteristics included in the weighting algorithm, this may produce unreliable estimates of population-weighted totals.”

Further, we have clarified how the US census data were used in combination with the ACS. We revised these details as follows (page 12): “We estimated population totals from the American Community Survey (ACS) Public Use Microdata Sample (PUMS) from 2009-2011.”

5. Finally, the point the authors make about the ABCD study sampling approach is relevant and timely. Of note, the current approach used by ABCD is focused on random sampling: <http://abcdstudy.org/school-selection.html>, though the study will need to stick to this approach to achieve the proposed sampling frame. The authors’ point is important as this study is the perfect example of a study that will be much more impactful and valuable if the sampling frame is clear. That said, they may want to at least acknowledge that the current goals of ABCD are to use random sampling at the school level.

We thank the reviewer for calling our attention to the current sampling approach proposed in the ABCD study. We have revised our discussion of this study to reflect this development (See page 9):

“...[W]e suggest the need for thoughtful sampling strategies that explicitly consider the target population of interest, and reflect an attempt to broadly recruit and represent such populations. An example of how this might be done in practice is observed in the National Institutes of Health recently funded Adolescent Brain and Cognitive Development (ABCD) Study, which aims to assess brain development in over 10,000 adolescents from across the country. The ABCD study investigators intend to recruit a sample of boys and girls that represent the U.S. population of adolescents, and have developed a school-based recruitment strategy to accomplish this goal (<http://abcdstudy.org/school-selection.html>). If conducted according to plan, this will be the first neuroimaging study that generates a true nationally representative sample, and will help ensure that the results of the study are generalizable to all adolescents in the U.S. We suggest that this and future large-scale neuroscience initiatives include collaboration with population scientists (i.e. epidemiologists, demographers) to ensure that these sampling concerns are adequately addressed over the long term.”

I applaud the authors for their efforts and hope this work can help push the field forward to a more generalizable and replicable understanding of human neuroscience. I appreciate the opportunity to contribute to reviewing this manuscript.

Thank you for your constructive and helpful comments!

Signed: Luke W. Hyde, University of Michigan

Reviewer #2 (Remarks to the Author):

In “The representative developing brain: Does sampling strategy matter for neuroscience,” LeWinn et al investigate the degree to which non-representative sampling may impact inference regarding normative patterns of brain development. This manuscript has a good deal to recommend it, including an important topic, generally clear writing, and use of a large dataset. However, while I found this manuscript interesting, I felt that it left some aspects of this important problem under-explored. Suggestions for the authors to consider follow below:

1. Unpacking of results. The authors show that fitted models differ after weighted samples were constructed, but do not explain what drove those effects? Were some demographic effects more important to consider than others?

We thank the reviewer for calling our attention to this issue. Our primary goal in this study was to empirically examine whether sample composition meaningfully impacts fundamental questions in cognitive neuroscience like the association between age and brain structure. Our intention was not to develop a hierarchy of what variables are most important to consider for the purposes of sample weighting. Even if this was our intention, the methods by which one would generate this hierarchy are unclear. We would need to include weighting variables individually and in combination to test which of these variables led to the greatest difference between unweighted and weighted models. Neither the raking procedure we employed nor other sample weighting methods provides statistics for measuring effect sizes associated with specific weighting variables. Furthermore, the strong links between race/ethnicity and socio-economic status in the U.S. would make it challenging to disentangle which of these factors is more important to represent well in one’s sample. However, even if there were a clear statistical procedure by which to create a hierarchy of characteristics most important to include in a weighting algorithm, we have several conceptual concerns with this suggestion that we explain below.

First, we included only basic socio-demographic characteristics in our weighting procedure. These characteristics were chosen because (1) their distributions are often very different in neuroimaging samples than in the U.S. population as a whole, (2) they were available in the PING dataset, and (3) they were available in the U.S. Census data, allowing us to identify the distribution in the U.S. population. These characteristics do not represent all the possible attributes of a sample that may influence the association between age and brain structure. As we mention in our paper and as further emphasized by Reviewer 1, there are likely many unmeasured characteristics (e.g. birth weight, exposure to toxins, urban/suburban versus rural neighborhood, exposure to stress and adversity) whose distributions in the PING dataset are very different than the distribution in the broader U.S. Therefore, without the ability to consider some of these additional characteristics in our weighting algorithm, we are reticent to use this data to make broader statements about which characteristics are most important to consider in a neuroimaging study.

Finally, even if we did have complete information on the PING participants and on the entire U.S. population, and a method by which to evaluate which variable was most influential in the weighting process, the characteristic that most influenced differences between the weighted and unweighted models would be specific to the PING dataset. In other words, a study with a different underlying distribution of these characteristics would likely yield a different conclusion about which aspects of sample composition are most relevant for the age-brain structure relationship. Furthermore, the influence of any given characteristic likely changes in relation to the hypothesis being tested (e.g. associations between brain structure and cognition may be more sensitive to the distribution of birth weight in the sample). Therefore, we believe that any attempts to generate a hierarchy of socio-demographic variables included in our weighting algorithm would be relevant only for our specific question and dataset, and would also detract from our overall message, which is that sample composition meaningfully impacts neuroimaging findings and future studies should include more well defined sampling strategies that better reflect the target population of interest.

2. Data quality. Multiple studies (Reueter et al., Neuroimage 2015; Pardoe et al. Neuroimage 2016; Ducharme et al., Neuroimage 2016) have shown the impact of data quality on measures derived from structural images. I

imagine that these concerns are as big if not bigger than those outlined in this paper—ignoring this factor in the weighting procedure seems like a potentially substantial flaw.

The PING study used online motion correction during scan acquisition to reduce motion artifacts and a rigorous post-processing quality control procedure for the T1 data that dropped subjects with significant head motion. Quality control procedures included visual inspection of the images by trained imaging technicians and computer algorithms testing general image characteristics, contrast properties, registrations, and artifacts from motion and other sources. Approximately 20% of subjects were dropped from analysis prior to exporting FreeSurfer statistics due to quality control issues, such as motion. Although residual artifact is certainly possible, the rigorous quality control procedures used in PING increase our confidence that artifact is not driving the results.

Quantitative analysis of how motion might contribute to the differences in the weighted and unweighted data is not possible, as the PING study did not publically release information on motion or artifact at the subject-level. We do not have information on brain metrics from subjects who were dropped due to artifact, precluding us from examining the degree to which differential artifact might contribute to differences across the weighted and unweighted samples. Nevertheless, the rigorous quality control procedures used by the PING study reduces concern about substantial artifact in the subjects included in our analysis.

Finally, evidence clearly indicates that motion produces reductions in estimates of cortical thickness and volume (Reuter et al., 2015). If differential motion were driving the effects between the weighted and unweighted samples, we would expect that motion would be higher in the weighted sample in which low SES children were weighted more heavily to address their under-representation in PING. This would produce a pattern of reduced cortical volume and thinner cortex in the weighted sample, particularly in the youngest children with the greatest head motion. However, we find the opposite pattern. In the weighted sample the youngest children have *greater* cortical volume than in the unweighted sample. This makes it unlikely that motion is explaining these findings. Motion is also unlikely to influence other findings, such as the curvilinear associations between age and volume of subcortical structures and cortical surface area. Given the profound and generally linear effect that age has on motion, these curvilinear associations are unlikely to be accounted for by variation in motion across these ages. Nonetheless, it is certainly possible that small motion artifacts contribute somewhat to the variability across the weighted and unweighted samples. We have added differential motion artifacts as an alternative interpretation of our findings in the revised manuscript on page 8:

“An alternative explanation of differences between the weighted and unweighted models is that they reflect differential motion artifact among lower-SES children who were weighted more heavily to address their under-representation in the unweighted sample. Although it is certainly possible that small motion artifacts contribute somewhat to the variability across the weighted and unweighted samples, motion is unlikely to entirely explain the observed differences for several reasons. First, the PING study used a real-time motion correction algorithm during data acquisition^{40,41} and rigorous quality control procedures for the T1 data that dropped subjects with significant head motion. Second, if differential motion were driving these effects, we would expect that motion would be higher in the weighted sample where low-SES children had greater representation. This would produce a pattern of reduced cortical volume and thinner cortex in the weighted sample, particularly in the youngest children, as greater motion is associated with reductions in cortical thickness and volume^{42,43}. However, we find the opposite pattern. In the weighted sample, the youngest children have greater cortical volume than the youngest children in the unweighted sample, indicating that motion is an unlikely explanation for these findings.”

Reuter, M., Tisdall, M. D., Qureshi, A., Buckner, R. L., van der Kouwe, A. J. W., & Fischl, B. (2015). Head motion during MRI acquisition reduces gray matter volume and thickness estimates. *NeuroImage*, 107, 107–115

3. Datasets. There are increasing numbers of large-scale developmental datasets available. While PING is a good choice for this study, I was surprised that the authors did not use (or even reference) the NKI-Rockland sample as a counterpoint to PING, as it was explicitly designed with representative sampling in mind. Showing that after weighting the PING dataset became more like the Rockland sample would increase confidence in the findings and provide a valuable replication.

The purpose of our analysis was to compare estimates of neural structure by age in a large sample of children and adolescents before and after the sample was weighted to be more representative of the United States. The goal was to determine—within the same sample, using the same pre-processing pipeline and modeling approach—whether sample composition would influence the association between age and brain structure. Comparison of the results of our weighted sample to the Rockland sample would not achieve this objective, as the Rockland sample is not representative of the United States and differs in several important ways from the US population. As a result, the Rockland sample is likely to differ in meaningful ways from our weighted sample.

Specifically, the NKI-Rockland sample was designed to be representative of a small geographical area within the United States (Rockland County, NY), not the United States as a whole. As a result, the demographic characteristics of the NKI-Rockland sample are different from those of the United States in significant ways. The NKI-Rockland sample has substantially higher levels of parental education (41% with a bachelor's degree compared to 28% in the United States) and parental income (median income in this sample is \$82,000 compared to \$52,000 in the United States). These differences in socioeconomic status (SES) between the Rockland sample and the United States are significant, because SES is strongly related to neural structure in children (Mackey et al., 2015; Noble et al., 2015). Moreover, the age distribution of the pediatric portion of the Rockland sample is different than in PING (i.e., the Rockland sample includes children age 6 years and older, compared to PING which includes children age 3 years and older) and includes far fewer children per age than in PING, meaning that the sample is under-powered to detect age-related variation, particularly at younger ages. As such, we do not agree with the following statement: “*showing that after weighting the PING dataset became more like the Rockland sample would increase confidence in the findings and provide a valuable replication.*” Because the Rockland sample is not representative of the U.S., it does not provide a benchmark for our study or other studies attempting to generalize findings to the population of the U.S. As noted above, the goal of our study was to approximate a U.S. representative sample by applying a robust sample weighting methodology to a very large community sample of children that included the full distribution of important socio-demographic characteristics such as race/ethnicity and SES. Furthermore, the PING study included individuals from multiple geographic regions across the United States. Because the Rockland sample was geographically isolated, we anticipate that the distribution of unmeasured characteristics that may be related to brain development (such as birth weight, prenatal exposures, and environmental exposures to toxins) is different in the Rockland sample than in the United States. Overall, we see little benefit of comparing our results to an analysis of the Rockland sample. This comparison would be uninterpretable, as the Rockland sample is not a nationally representative sample, does not include children under six, and does not have a large number of children within each age group.

While not a suitable comparison for our study, we do appreciate the efforts of the NKI Rockland investigators to generate a representative sample of their county, and explicitly mention this study as a laudable example of prior imaging work that has attended to population science principles (See page 3, paragraph 1).

Mackey, A. P., Finn, A. S., Leonard, J. A., Jacoby-Senghor, D. S., West, M. R., Gabrieli, C. F. O., & Gabrieli, J. D. E. (2015). Neuroanatomical Correlates of the Income-Achievement Gap. *Psychological Science*, 26(6),

Noble, K. G., Houston, S. M., Brito, N. H., Bartsch, H., Kan, E., Kuperman, J. M., et al. (2015). Family income, parental education and brain structure in children and adolescents. *Nature Neuroscience*, 18(5), 773–778.

4. *Site effects. PING was a multi-site study, and site effects can be prominent in imaging data. Did the authors evaluate or control for these effects in their models?*

We did include site as a covariate in our analysis; however, we see how our original description of this covariate was unclear and thank the reviewer for calling our attention to this issue. We have edited this section in our methods to be clearer (page 12):

“All models included covariates for sex, race/ethnicity, parent educational attainment, family income, and scanner site.”

5. *Modeling. I was surprised the authors used polynomial models selected by AIC when the field has moved to*

data-driven semi-parametric models, such as general additive models; the PING consortium and others use primarily use this approach. I wonder if GAMs were to be used whether the same results would be found? Furthermore, rather than using low-dimensional summary measures like time at peak, I might prefer a quantitative model comparison over the entire age range—i.e., are the fitted values different only at peak, or at multiple age ranges?

The goal of our study was to demonstrate the potential influence of sample composition on age-related variation in global measures of brain structure. In order to test this question, we compared models of age-related variation in the original PING sample (i.e., the unweighted sample) to the PING sample after we applied sample weights to make the distribution of several socio-demographic factors equivalent to the distribution in the broader United States population. We applied a parametric modeling strategy that allowed us to choose the best fitting models for the unweighted and weighted sample by using statistical tests of model fit (i.e. magnitude of effects, standard errors of polynomial terms and AIC). This approach was important for our demonstration because it (1) overlaps with standard modeling approaches used in developmental cognitive neuroscience (Ostby et al., 2009; Shaw et al., 2008, Tamnes et al., 2010; Mills et al., 2016), and (2) allowed us to quantitatively compare the best fitting polynomial terms (e.g. linear, quadratic, cubic) across the unweighted and weighted samples. This approach allowed us to determine—using the same criteria—whether the best-fitting model was different in the weighted sample as compared to the unweighted sample.

The challenge with using general additive models, which are non-parametric, is that it is not possible to use standard statistical approaches for estimating and comparing model fit. In other words, if we estimated a general additive model, it is not possible to determine whether that model fits the data better than a parametric model with linear, quadratic, or cubic terms. This makes it impossible to determine the best-fitting model using a quantitative approach. Our analytic approach relies on using identical procedures for determining the best-fitting model in the weighted and unweighted samples before comparing them to one another. GAMs would not allow us to use a quantitative approach for identifying the best-fitting model in each of our samples, making it challenging to conclude with certainty that the answer arrived upon would be different if the sample composition were different.

More broadly, we think it's important to point out that the goal of our study was to determine whether sample composition has a meaningful effect on age-related variation in global measures of brain structure. The goal was not to arrive at the “most definitive” or “final” answer of how the brain changes with age. Instead, our goal was to demonstrate that the answer one arrives at regarding how brain structure varies with age using approaches common in cognitive neuroscience differs depending on the composition of one's sample. Thus, we are not arguing that the weighted data presented in the paper is the final answer or the best answer to the question about how brain structure varies with age. Instead, we are arguing that the answer looks different in fairly meaningful ways depending on the composition of one's sample. The more general conclusion is that investigators must be cautious to interpret the findings from neuroimaging samples in the context of their sample. Prior work from PING and other groups regarding age-related variation in brain structure is not incorrect – it is reflective of age-related variation in the particular sample in question, but not necessarily age-related variation in the broader population. We demonstrate here that our conclusions about age-related variation in brain structure might be different if samples more closely resembled the U.S. population.

Nonetheless, we acknowledge that there are other methods, such as GAMs, that may more flexibly model age-related variation in brain structure would be beneficial to apply in future studies. We mention in the revised paper that there are evolving statistical approaches to examining developmental variation that may better capture non-linear patterns across time than the approaches used here, but that we were limited to statistical methods that allowed us to directly compare the best-fitting model from our weighted and unweighted samples (see page 9 paragraph 1).

Ostby, Y., Tamnes, C. K., Fjell, A. M., Westlye, L. T., Due-Tønnessen, P., & Walhovd, K. B. (2009).

Heterogeneity in subcortical brain development: A structural magnetic resonance imaging study of brain maturation from 8 to 30 years. *Journal of Neuroscience*(29), 11772–11782.

Shaw, P., Kabani, N. J., Lerch, J. P., Eckstrand, K., Lenroot, R. K., Gogtay, N., Greenstein, D., Clasen, L., Evans, A., Rapoport, J. L., Giedd, J. N., & Wise, S. P. (2008). Neurodevelopmental trajectories of the human cerebral cortex. *Journal of Neuroscience*, 28, 3586-3594.

- Tamnes, C. K., Ostby, Y., Fjell, A. M., Westlye, L. T., Due-Tønnessen, P., & Walhovd, K. B. (2010). Brain maturation in adolescence and young adulthood: regional age-related changes in cortical thickness and white matter volume and microstructure. *Cerebral Cortex*, 20(3), 534–548.
- Mills, K. L., Goddings, A.-L., Herting, M. M., Meuwese, R., Blakemore, S.-J., Crone, E. A., et al. (2016). Structural brain development between childhood and adulthood: Convergence across four longitudinal samples. *NeuroImage*, 141, 273–281.

Reviewer #3 (Remarks to the Author):

The paper examines the issue of non-representation in imaging studies in the neurocognitive literature. The authors find that there are some aspects of imaging that find little differences in a weighted, representative sample but in regional areas some quite important differences. The authors offer a novel use of weights to examine the potential bias that imaging studies may have due to lack of representing the population. This paper gives some evidence that the biases can be important and substantial.

Thank you for these positive comments.

The article is convincing. The only real weakness, and this is addressed by the authors, is that this study is not representative to begin with and so the estimates are of the potential bias. Also, the authors made reference to the ABCD 10,000 adolescent study and that study is using population sampling to ascertain the sample. That will be one of the first to really have the representation of both the brain and biology and will be available for researchers across the world to use.

Thank you for pointing this out. We have expanded our discussion of the ABCD study in the revised discussion section to highlight the issues the Reviewer points out (see page 10):

“An example of how [representative sampling in neuroimaging] might be done in practice is observed in the National Institutes of Health recently funded Adolescent Brain and Cognitive Development (ABCD) Study, which aims to assess brain development in over 10,000 adolescents from across the country. The ABCD study investigators intend to recruit a sample of boys and girls that represent the U.S. population of adolescents, and have developed a school-based recruitment strategy to accomplish this goal (<http://abcdstudy.org/school-selection.html>). If conducted according to plan, this will be the first neuroimaging study that generates a true nationally representative sample, and will help ensure that the results of the study are generalizable to all adolescents in the U.S.”

In addition, we agree that because the PING sample was not representative to begin with, there are numerous unmeasured characteristics that may still differ in the weighted sample as compared to the broader U.S. population (e.g., birthweight, urban vs. rural residence, exposure to toxins). Because these characteristics were unmeasured in the PING sample, we could not weight on them. We have noted this limitation in the revised discussion section (see page 8):

“In a true representative sample of children, the distribution of unmeasured characteristics would, on average, reflect that of the U.S. Our sample weights account for distributions of the measured characteristics of sex, race/ethnicity, parental education, and income, but not for all unmeasured characteristics that would render a sample representative of a target population. Characteristics that may further influence the age-brain structure association include birth weight³⁸, exposure to prenatal toxins, and exposure to traumatic violence³⁹; however, these characteristics were not weighted in the present study. Less frequently studied characteristics may also influence age-brain structure relationships. For example, PING participants were recruited from nine urban study sites, leading to greater representation of children from urban and suburban areas and under-representation of children from rural locations. Future studies unable to implement a random sampling procedure but interested in post-stratification weighting may choose to assess additional participant characteristics (e.g. urban, suburban, rural) and use weighting techniques that include more complex algorithms to account for sample characteristics beyond basic socio-demographics. However, there is no substitute for true, representative sampling as traits endogenous to the participants that are too numerous to measure comprehensively may influence the likelihood of participation in samples of convenience (e.g. some

participants may be more likely to hear of and participate in the study by word of mouth than others as a result of their social networks). As a result, although the weighted PING sample more closely resembles the U.S. population than the unweighted sample, it is not a truly representative sample.”

The statistical analyses appears appropriate and relies on looking at differences in functional forms vs. mean differences. The reviewers have provided detail of their method and a link to their study and are providing the data. Thus, it should be straight forward to perform a replication of the data given the openness of the data and the procedures used in this study.

Thank you for drawing our attention to this issue. We plan to include our statistical code used for the raking procedure with the final manuscript to allow other investigators to use this approach if they have samples with sufficient power and distribution of socio-demographic characteristics to use sample weighting techniques.

Overall, a good examination of the issue of lack of representation of neurocognitive studies in the field. It is also a call to action to consider the issue of universal processes when almost no studies have actually taken into account important differences in college samples and population samples.

Thank you for these positive comments.

Pamela Davis-Kean

Reviewer #4 (Remarks to the Author):

LeWinn and colleagues describe the impact of “representativeness” (or a lack off) on estimating trajectories of brain development in community-based samples.

This is a well-written manuscript with a very important message. I have only a few minor comments and suggestions.

In the second paragraph (around line 70 and line 83), you may wish to point out a census-based approach employed in the NIH MRI Study of Normal Brain Development. This is one of a few studies that attempted to do what the authors are calling (visi-a-vis demographic variables) for when recruiting children into an imaging study. The same study, however, introduced other biases – for example an extremely long list of exclusion criteria that likely resulted in a “supernormal” sample...

We agree. This study is cited in the paper and we have noted it by name in the revised manuscript (see page 3):

“In the NIH MRI Study of Normal Brain Development, a foundational study upon which significant knowledge of structural brain development is based, investigators selected a community-based sample representative of the population in the study areas¹⁶; however, this study included numerous exclusion criteria (e.g., the presence of clinically-significant mental health symptoms) that reduced the true representativeness of the sample¹⁷.”

On line 231, you may wish to add “created” after “children” to avoid a misunderstanding that this sample was representative to start with...

Thank you, we have better clarified this sentence to indicate that our weighted sample was not representative (see page 6):

“To address this gap in the literature, we approximated a representative sample of U.S. children by applying a commonly used epidemiologic method of sample weighting to a large, community-based sample of typically developing children and estimated associations of age with global and regional measures of grey matter brain structure in this weighted sample.”

(We address the following two comments by this reviewer in a single response below.)

Around line 111, you may wish to draw attention to a recent paper by Ducharme (PMID:26463175) suggesting that non-linear relationships between cortical thickness and age may be “induced” by head motion. Please see our expanded response to issues concerning head motion below. We have cited the Ducharme paper you suggest in the revision.

In the paragraph about the most pronounced differences between unweighted and weighted models in subcortical volumes (lines 288-300), you may wish to consider an alternative explanation: subcortical volumes are more difficult to estimate (than cortical volumes) and, possibly, more sensitive to imaging artifacts (such as head motion – see the Ducharme paper mentioned above). Could this be why the two models differ most?

We thank the reviewer for these salient points related to the potential impact of head motion on our findings and to calling our attention to the Ducharme paper. If differential motion were driving differences between unweighted and weighted samples, we would expect that motion would be lower in the unweighted sample where low SES children had lower representation. This would produce a pattern of greater cortical volume and thicker cortex in the unweighted sample, particularly in the youngest children. In the weighted sample, we would expect reductions in cortical volume and thickness, as greater motion is associated with reductions in cortical thickness and volume (Ducharme et al., 2016; Reuter et al., 2015). However, we find the opposite pattern. In the weighted sample the youngest children have greater cortical volume than in the unweighted sample. This makes it unlikely that motion is explaining these findings. Motion is also unlikely to influence other findings, such as the curvilinear associations between age and volume for subcortical structures and age and surface area for cortical structures. Given the profound and generally linear effect that age has on motion, these curvilinear associations are unlikely to be accounted for by variation in motion across these ages. Nonetheless, it is certainly possible that small motion artifacts contribute somewhat to the variability across the weighted and unweighted samples.

With regard to subcortical structures, it is possible that the differences between the weighted and unweighted models are due to the fact that these regions are more challenging to image (and thus have greater measurement error than larger cortical regions) or because they are more susceptible to head motion. Susceptibility of these regions to motion artifact would have to be substantially different between the weighted and unweighted models for this to be a reasonable explanation. Although this is possible, the PING study used a real-time motion correction algorithm (PROMO) during data acquisition and rigorous quality control procedure for the T1 data that dropped subjects with significant head motion. This allays concerns somewhat that differences between the samples simply reflect differences in motion artifact. Furthermore, weighting had a greater impact on age-related variation not only in sub-cortical structures but also in cortical lobe surface area as compared to total cortical surface area. Thus, it is not only in sub-cortical structures that we observe meaningful effects of weighting.

Ducharme, S., Albaugh, M. D., Nguyen, T. V., Hudziak, J. J., Mateos-Pérez, J. M., Labbe, A., ... & Brain Development Cooperative Group. (2016). Trajectories of cortical thickness maturation in normal brain development—The importance of quality control procedures. *NeuroImage*, 125, 267-279.

Reuter, M., Tisdall, M. D., Qureshi, A., Buckner, R. L., van der Kouwe, A. J. W., & Fischl, B. (2015). Head motion during MRI acquisition reduces gray matter volume and thickness estimates. *NeuroImage*, 107, 107–115.

We have added the following paragraph to our limitations section to address these concerns about possible motion artifacts:

“An alternative explanation of differences between the weighted and unweighted models is that they reflect differential motion artifact among lower-SES children who were weighted more heavily to address their under-representation in the unweighted sample. Although it is certainly possible that small motion artifacts contribute somewhat to the variability across the weighted and unweighted samples, motion is unlikely to entirely explain the observed differences for several reasons. First, the PING study used a real-time motion correction algorithm during data acquisition^{40,41} and rigorous quality control procedures for the T1 data that dropped subjects with significant head motion. Second, if differential motion were driving these effects, we would expect

that motion would be higher in the weighted sample where low-SES children had greater representation. This would produce a pattern of reduced cortical volume and thinner cortex in the weighted sample, particularly in the youngest children, as greater motion is associated with reductions in cortical thickness and volume^{42,43}. However, we find the opposite pattern. In the weighted sample, the youngest children have greater cortical volume than the youngest children in the unweighted sample, indicating that motion is an unlikely explanation for these findings.”

When talking about calls “to incorporate population science approaches into neuroimaging research (line 48) and “into cognitive neuroscience” (line 228), you may wish to cite Paus 2010 (PMID:20496380) and 2013 (ISBN 978-3-642-36450-1).

Thank you for this suggestion – we have cited these papers in the revised manuscript (see page 6).

Reviewers' comments:

Reviewer #1 (Remarks to the Author):

The authors have been thoughtful and responsible in their revisions. I agree with their responses to my reviews/suggestions and to the points of the other reviewers. I believe their responses are a boon to the field as they help to clarify the strength and novelty of their approach and the impact on the field. I have no other comments and commend the authors on an exciting and important study.

Reviewer #2 (Remarks to the Author):

In their response to the initial review, Le Winn et al provide a detailed rebuttal to all comments offered. Sadly, I did not find their arguments altogether convincing, and together they markedly diminish my enthusiasm for the manuscript. Taken together, while this study is not without value, overall it seems more appropriate for a specialty journal rather than Nature Communications. This opinion is primarily driven by two points which seem to be major unresolved weaknesses of the manuscript:

1) Unpacking of results. If the authors cannot describe what drives their observed results, then it renders interpretation much more difficult and less meaningful. The authors claim that there is no way to help further unpack their results given the method used; that strikes me as a major limitation of the method, and reduces interest in the findings.

2) Datasets. Given the high impact of the target journal, and the general nature of the analyses, it is reasonable to expect a replication sample. The authors did not acquire any new data or develop a new method; given that this procedure could be easily replicated with additional public datasets, I am somewhat confused why they did not reproduce their results. Replication would increase confidence that results are not specific to this particular dataset. I mentioned the NKI dataset earlier, but other options include the Philadelphia Neurodevelopmental Cohort and the NIMH intramural study of brain development. Even stronger would be to acquire additional data prospectively on a population-matched sample as a "negative control" analysis.

Reviewer #3 (Remarks to the Author):

I have reviewed the revised manuscript, the comments from the authors to the reviewers comments, and the supplemental materials. I found that the manuscript has been significantly enhanced and that the authors have done an excellent job in responding to the reviewers comments.

I have no additional comments or recommendations for changes for the authors.

Reviewer #4 (Remarks to the Author):

All my comments have been addressed. One minor suggestions - the authors may replace "subjects" by "participants" throughout the text.

RESPONSE TO REVIEWERS' COMMENTS

We thank the reviewers and the Nature Communications Editorial team for their thoughtful comments on our revised manuscript. We were pleased to see the enthusiasm for our paper from the reviewers and the editorial staff. Given the detailed feedback and the specific requests we received from the editorial team in this round of reviews, which also encompassed the remaining concerns from Reviewer 2, we focus our response to reviewers on these editorial comments.

Editorial Comments:

We interpreted the primary concerns with our manuscript to be summarized in the following paragraph: *While we recognize that this is a descriptive study, we ask that you provide detailed numerical descriptions and statistical comparisons of your results and also that you list all formulas used in the Methods section. Please be sure to accompany every generally descriptive claim (e.g. larger or smaller) with exact numbers specifying the size and the difference between them, and whenever possible, please use the appropriate statistical test to determine whether the observed difference is significant or not. Please also report the effect sizes, regardless of whether the p value is significant or not (for further details, see attached manuscript).*

In the revised version of our manuscript we have included numerical descriptions and detailed statistical tests whenever possible. We also clearly articulate throughout the results section and in the methods which results are compared with statistical tests and which are provided to aid in the interpretation of our findings.

To briefly summarize, our primary goal in this study was to empirically examine whether sample composition meaningfully impacts fundamental questions in cognitive neuroscience like the association between age and brain structure. To accomplish this, we compared the best fitting models for the age-brain structure associations in unweighted data from a community based sample and data from that same community sample weighted to look more like the U.S. population on basic sociodemographic characteristics. The primary statistical and quantitative tests are those we used to determine the best fitting regression models for the association between age and brain structure in the unweighted and weighted data. For each association, we determined whether a linear, quadratic, or cubic term for age provided the best fit to the data by comparing the Akaike Information Criteria (AIC), which provides a quantitative summary assessing how well a statistical model aligns with the underlying data compared with other models of the same data. AIC is commonly used for model selection (i.e., selecting covariates that provide the best fit to the data and selecting the best functional form of a model), and is a superior approach for determining model fit than simply determining whether additional parameter estimates are statistically significant (Bozdogan, 1987; Buckland, Burnham, & Augustin, 1997; Sclove, 1987). Model fit statistics determine how well a particular model aligns with the underlying data, while taking into account the number of parameters in that model (rather than examining the statistical significance of each parameter individually). Model fit has long been accepted as the gold standard approach for model selection across a wide range of

scientific disciplines, including the behavioral sciences and epidemiology (Burnham & Anderson, 2003; Sclove, 1987); this approach is particularly well-suited for deciding among models with polynomial terms (Sclove, 1987), as we do in the current report. Note that the Burnham & Anderson (2003) book on model selection, which emphasizes model fit statistics as essential before interpreting the statistical significance of parameter estimates, has been cited over 36,000 times.

To this revised version, we have added a new table (Table 2) that describes the beta estimates, standard errors, and AIC values for each model we tested in both the unweighted and weighted data. Throughout the manuscript, we were asked to include effect estimates with each comparison we made between unweighted and weighted models. To preserve the readability of our manuscript, we explain in detail the results for total cortical surface area but refer readers to Table 2 for other brain structures.

Once we determined the best fitting regression model for age and each measurement of brain structure in both the unweighted and weighted data, we generated graphs of the predicted values for each brain structure metric by age. These results were included primarily to aid in the interpretation of model fit differences, as it is challenging to visualize the shape of a quadratic or cubic trajectory from beta estimates alone and some quadratic and cubic models can describe trajectories that are effectively linear (which we observed in our cortical thickness measurements). However, because these graphs were generated from independent datasets (i.e., weighted and unweighted), we cannot statistically compare aspects of those graphs (e.g. differences in slopes between the unweighted and weighted models). Throughout the paper, we have carefully explicated the descriptive purpose of these graphs. Please note that graphing predicted values to assist in interpretation of quadratic and cubic effects in developmental studies of brain structure is common practice (Gogtay et al., 2004; Shaw et al., 2008).

Similarly, age at peak surface area and volume were also calculated to aid in the interpretation of our results; however, similar to the predicted value graphs, we cannot statistically compare peak ages between unweighted and weighted models because they are derived from independent samples. Age at peak surface area and volume are metrics commonly calculated in developmental cognitive neuroscience, and have been used, for example, to compare gender specific rates of development (Wierenga et al., 2014), as well as to identify basic developmental patterns (Gogtay et al., 2004; Mills et al., 2016). We apply the same methods for calculating age at peak volume and area as in prior work (i.e. by calculating the first order derivative for quadratic and cubic models). While we cannot claim that the differences in age at peak surface area or volume summarized in Table 3 are statistically and significantly different between the unweighted and weighted data, the inclusion of these metrics further demonstrate the potential impact of sample composition. For example, in a recent study of the relative timing of age at peak surface area across 84 regions of interest (ROI) among youth aged 7 to 23, authors observed that the earliest maturing regions reached peak surface area at age 8, compared to the latest maturing regions which reached peak area at age 11 (i.e. 3 years difference between the earliest and latest maturing regions across the entire brain) (Wierenga et al., 2014). In our study, we find differences in age at peak

surface area *for the same structure* that are over 3 years earlier in the weighted models compared to the unweighted models. This suggests that sample composition has an effect that is quite meaningful. We include age at peak surface area and volume to demonstrate the potential impact of sample composition on metrics that are commonly derived from best fitting regression models in developmental cognitive neuroscience and include a reference to the Wierenga study in our discussion of these results to clarify this point (See Discussion page 8).

Throughout the manuscript, we have included quantitative results for each finding we summarize in the text (which may include referencing readers to Table 2 in our summary of the best fitting models). We also moved all results except for those for intracranial volume into the main manuscript, which now includes all figures for cortical thickness predicted values (see Figure 3).

We have included all relevant formulas into our manuscript to fully explicate both the sample weighting procedure we applied as well as the first order derivative calculation we used to estimate age at peak surface area or volume (see pages 15 and 16 of the methods section). In conducting this careful review of our manuscript, we noted a few minor errors in our age at peak surface area calculations in what is now Table 3, which have been corrected. This correction did not alter any of the interpretations or conclusions of the manuscript.

Finally, several additional changes were made to address formatting requirements of the journal. For example, we removed subheadings within the Discussion and also moved some parts of the discussion to Supplementary Information to better comply with word count requirements. Please note that, as a result of addressing prior reviewer comments and making the editorial changes requested in our results section in this version, our manuscript word count is slightly above the 5,000 words suggested for the Introduction, Results, and Discussion. We are happy to work with the editorial staff to address this issue if necessary.

Addressing Comments within the Manuscript

Throughout the manuscript, we were asked to make several copy-editing corrections (e.g. include one level of subheadings in the Results section). We have made those changes as requested. We have also moved all but one figure and table into the main paper as requested. Finally, we were asked to provide a much more detailed description of the neuroimaging methods applied to the PING dataset. We have substantially revised this section and have included additional references to describe these methods (see pages 13-14 in the Methods).

Bozdogan, H. (1987). Model selection and Akaike's information criterion (AIC): The general theory and its analytical extensions. *Psychometrika*, 52, 345-370.

Buckland, S. T., Burnham, K. P., & Augustin, N. H. (1997). Model selection: an integral part of inference. *Biometrics*, 53, 603-618.

- Burnham, K. P., & Anderson, D. R. (2003). *Model selection and multimodel inference: a practical information-theoretic approach*. Fort Collins, CO: Springer.
- Gogtay, N., Giedd, J., Lusk, L., Hayashi, K. M., Greenstein, D., Vaituzis, A. C., Nugent, T. F., Herman, D. H., Clasen, L., Toga, A. W., Rapoport, J. L., & Thompson, P. M. (2004). Dynamic mapping of human cortical development during childhood through early adulthood. *Proceedings of the National Academy of Sciences*, *101*, 8174-8179.
- Mills, K. L., Goddings, A.-L., Herting, M. M., Meuwese, R., Blakemore, S. J., Crone, E. A., Dahl, R. E., Guroglu, B., Raznahan, A., Sowell, E. R., & Tamnes, C. K. (2016). Structural brain development between childhood and adulthood: Convergence across four longitudinal samples. *Neuroimage*, *141*, 273-281.
- Sclove, S. L. (1987). Application of model-selection criteria to some problems in multivariate analysis. *Psychometrika*, *52*, 333-343.
- Shaw, P., Kabani, N. J., Lerch, J. P., Eckstrand, K., Lenroot, R. K., Gogtay, N., Greenstein, D., Clasen, L., Evans, A., Rapoport, J. L., Giedd, J. N., & Wise, S. P. (2008). Neurodevelopmental trajectories of the human cerebral cortex. *Journal of Neuroscience*, *28*, 3586-3594.
- Wierenga, L. M., Langen, M., Oranje, B., & Durston, S. (2014). Unique developmental trajectories of cortical thickness and surface area. *NeuroImage*, *87*, 120-126 (2014)